# Simplifying Optimal Transport through Schatten-$p$ Regularization

**Tyler Maunu**                                    *maunu@brandeis.edu*
*Department of Mathematics*
*Brandeis University*

**Reviewed on OpenReview:** *https://openreview.net/forum?id=DIawkTG5VH*

## Abstract

We propose a new general framework for recovering low-rank structure in optimal transport using Schatten-$p$ norm regularization. Our approach extends existing methods that promote sparse and interpretable transport maps or plans, while providing a unified and principled family of convex programs that encourage low-dimensional structure. The convexity of our formulation enables direct theoretical analysis: we derive optimality conditions and prove recovery guarantees for low-rank couplings, barycentric displacements, and cross-covariances in simplified settings. To efficiently solve the proposed program, we develop a mirror descent algorithm with convergence guarantees in the convex setting. Experiments on synthetic and real data demonstrate the method's efficiency, scalability, and ability to recover low-rank transport structures. In particular, we demonstrate its utility on a machine-learning task in learning transport between high-dimensional cell perturbations for biological applications. All code is publicly available at `https://github.com/twmaunu/schatten_ot`.

## 1 Introduction

Optimal transport (OT) has emerged as a fundamental computational tool across many areas, including machine learning, computer vision, statistics, and biology (Arjovsky et al., 2017; Peyré and Cuturi, 2019; Schiebinger et al., 2019; Bonneel and Digne, 2023). It provides a principled framework for comparing probability distributions, and it has a rich mathematical history (Villani, 2008). While the combination of practical utility and deep mathematical theory has led to the broad adoption of OT ideas in mathematics, science, and engineering, finding ways to *scale* OT solutions and make them *interpretable* remains a fundamental research question (Cuturi et al., 2023; Khamis et al., 2024). In particular, OT typically suffers from the curse of dimensionality (Chewi et al., 2025), and regularized estimators may lack sparsity (Genevay et al., 2019).

A long line of work has focused on making OT scalable and interpretable through *regularization*. The most classical of these is entropic regularization, which yields a strictly convex program that can be solved via Sinkhorn scaling (Sinkhorn, 1967; Cuturi, 2013). More recent work has sought to increase efficiency and interpretability through quadratic regularization (Blondel et al., 2018; Lorenz et al., 2021), as well as low-rank factorizations (Forrow et al., 2019; Scetbon et al., 2021). These methods show promise in biological applications, particularly in single-cell RNA sequencing analysis (Klein et al., 2025).

Another closely related set of recent works attempts to include sparsity in the OT map using *elastic costs* (Cuturi et al., 2023; Klein et al., 2024; Chen et al., 2025). In these works, using different cost modifications can be shown to encourage sparse or low-rank transport displacements. This leads to OT maps with simple, interpretable structures.

Except for entropic regularization, our work simultaneously generalizes all of the aforementioned methods in a unified framework. We believe that this unified picture can lead to more principled development of tailored regularization. Furthermore, the theory of OT has not yet fully leveraged the extensive literature

on regularization for scaling and interpretability present in other fields, such as *compressed sensing*. In compressed sensing, the use of $\ell_1$ or nuclear-norm penalties as proxies for rank minimization has yielded provably efficient algorithms (Eldar and Kutyniok, 2012; Wright and Ma, 2022). Our general formulation marries ideas from OT and compressed sensing, providing a bridge that we expect to be fruitful for developing sparse and low-rank optimal-transport models moving forward.

## 1.1 Contributions

In this work, we present Schatten-$p$ regularized OT, which we call *Schatten OT*. This novel formulation is both general and amenable to direct theoretical analysis. We summarize the main contributions of our work:

- We demonstrate how the Schatten OT program simultaneously generalizes a large portion of prior work on low-rank and sparse methods in OT, while also yielding new regularized formulations.

- We propose a general mirror descent framework that efficiently solves the Schatten OT problem. Since the problems we consider are convex, this algorithm is guaranteed to converge to the global solution.

- Since our resulting optimization problems are convex, they allow analysis of the recovered transport plans. In particular, we theoretically demonstrate recovery of low-rank couplings, low-rank transport displacements, and low-rank covariance structures in toy settings.

- Experiments on synthetic and real data demonstrate the flexibility and effectiveness of Schatten OT in reducing dimensionality while at the same time balancing low transport cost.

OT has broad applications in machine learning. For example, in applied workflows, OT can be used to align distributions across domains, time points, or modalities. The fundamental point of our work is to introduce low-dimensionality into OT couplings in a principled and effective way. Low-dimensionality is important in practice since it can reduce memory and computational cost while simultaneously improving the interpretability of the transport structure. This is crucial for downstream tasks like label transfer in single-cell data or domain adaptation using feature embeddings. By making these biases convex, Schatten OT can be effectively estimated and analyzed. A final advantage of our framework is that it also unifies many prior regularizers (see Section 2.3). From an applications perspective, Schatten OT can be swapped in wherever practitioners currently use regularized OT and wish to encourage low-dimensional structure.

## 1.2 Related Work

Regularized variants of OT have become increasingly important in current applied and theoretical research. The story of regularized OT begins with entropic regularization (Cuturi, 2013), which has roots in Schrödinger (1932). More recent regularizations include quadratic and sparse regularization (Blondel et al., 2018; Lorenz et al., 2021; González-Sanz and Nutz, 2024), which seek to encourage sparse structures in the transport plan.

Other work has studied low-rank factorizations in couplings to scale OT. Forrow et al. (2019) define a notion of factored couplings. Scetbon et al. (2021) use this notion of low-rank factorization of the coupling to develop an efficient Sinkhorn algorithm for factored couplings. Later, Lin et al. (2021) use multiple couplings to move through anchor points. Halmos et al. (2024) propose a new algorithm to optimize over the LC factorization, and Halmos et al. (2025) use hierarchical low-rank structures.

Another line of recent work has studied the regularization of displacements. Cuturi et al. (2023) introduce the notion of elastic OT costs and show how to construct maps with sparse or low-rank structure. Later, Klein et al. (2024) introduce learnable parameters into these costs, enabling greater flexibility in selecting the regularizer. Chen et al. (2025) use neural networks to learn maps in these settings.

We note that the incorporation of low-dimensional structure in OT displacements dates back to earlier subspace-robust notions of OT. Paty and Cuturi (2019) compute Wasserstein distances over worst-case

subspaces in the ambient space. These methods have some practical statistical advantages (Niles-Weed and Rigollet, 2022).

We can broadly think of regularizing OT as encoding bias in the transport plan. However, there are many other ways the OT problem can be biased. For example, some works seek to encode biases by optimizing the ground cost used within OT. Alvarez-Melis et al. (2019) learn an OT with invariances using an alternating minimization procedure, and focus on optimization over Schatten-$p$ balls. Sebbouh et al. (2024) learn a matrix $M$ that defines an inner product cost between measures on different spaces. Jin et al. (2021) match distributions in different spaces using separate linear transformations. We note that these works implicitly regularize transport, as in subspace-robust OT.

In seeking a principled way to regularize and scale OT, we draw connections with *compressed sensing*. Compressed sensing focuses on recovering sparse structures from data. Original foundational works concentrate on recovering sparse vectors using $\ell_1$ regularization (Donoho, 2006; Candès et al., 2006). These ideas were later extended to low-rank matrices (Fazel et al., 2008), which used nuclear norm regularization. The use of more general Schatten-$p$ norms followed this (Nie et al., 2012). The extension of these regularizations to other settings has been fruitful. For example, Scarvelis and Solomon (2024) use it in the context of deep learning.

The ideas of compressed sensing are seeing a resurgence in the age of modern machine learning and AI. Sparse autoencoders have become a primary tool for practitioners studying mechanistic interpretability (Huben et al., 2024). Sparse coding and rate reduction form a recent framework for training deep models to develop "white-box" methods (Yu et al., 2020; 2023). Compression as a general technique is effective at demonstrating intelligence in simple puzzles (Liao and Gu, 2025). These examples show the importance and practicality of developing theoretically principled compression techniques for machine learning and AI problems.

### 1.3   On the Low-Rank Assumption

Schatten OT is most useful when the transport between two high-dimensional distributions is controlled by a small number of latent degrees of freedom. Importantly, in our framework, the Schatten penalty is applied to an *affine image* $\mathcal{A}(P)$ of the coupling $P$, so "low rank" should be interpreted relative to the chosen transport object $\mathcal{A}(P)$ rather than as a blanket property of OT solutions. In our examples, depending on the affine map $\mathcal{A}$, low-dimensional structure may appear as: (i) a coupling that admits a low-rank factorization, (ii) a barycentric map (or displacement) that is well-approximated by a low-rank operator, or (iii) low-rank second-order summaries such as a transport covariance. As a concrete motivating example, if the coupling $P$ has rank $r$, then it can be written as a sum of $r$ rank-one factors, which can be interpreted as $r$ transport modes (Forrow et al., 2019). Our method can be considered a convex analog of this past work.

**Single-cell and high-content perturbation studies.**   In single-cell perturbation experiments, each condition can be viewed as an empirical distribution in a high-dimensional feature space (e.g., gene expression, metabolic expression, protein markers, or learned image embeddings) (Schiebinger et al., 2019; Bunne et al., 2023). A low-dimensional assumption is most plausible when interventions act through a limited number of latent factors, so that the condition-to-condition shift is largely explained by a few coherent directions. In this setting, promoting low rank in the coupling or in the induced barycentric displacement yields a small number of transport archetypes that summarize how subpopulations move under perturbation, providing a compact object that can be inspected and compared across interventions.

**Imaging and registration.**   Optimal transport has many applications in imaging (Haker et al., 2004; Ferradans et al., 2014; Solomon et al., 2015). In image-based problems like registration, domain alignment, or matching related modalities, transport structure is more likely to be low-dimensional when deformations are coherent and can be parameterized by a small number of modes. When this holds, encouraging a low-rank coupling or barycentric map promotes alignments that reflect a few dominant deformation directions rather than a fully dense plan, while still controlling transport cost.

**Generative modeling and low-dimensional data manifolds.** Wasserstein-based machine learning has been effective in a variety of practical settings (Arjovsky et al., 2017; Courty et al., 2016). A low-dimensional bias is most natural when high-dimensional observations concentrate near a lower-dimensional manifold and the generator itself maps from a low-dimensional latent space to the ambient space. In such cases, the relevant transport between model and data distributions may be well-approximated by a locally low-dimensional mapping. Schatten OT provides a direct way to encode this bias at the level of the transport solution (e.g., via low-rank barycentric maps or transport covariances), yielding an explicit complexity–fidelity tradeoff in OT subroutines used within learning pipelines.

**Diagnostic indicators in practice.** While there are many applications where a low-rank structure is well-motivated, one should verify that this is indeed a good inductive bias when encountering a practical problem. A practical diagnostic for low-dimensional structure is spectral decay of the transport object of interest, summarized by effective-rank style measures (Roy and Vetterli, 2007). Empirically, the low-dimensional regime is suggested when principal components or other low-rank summaries capture a large fraction of variance within and across conditions. In practice, tuning the regularization parameter and monitoring how the estimated rank/complexity decreases as the transport cost increases can be an effective tool for assessing whether or not the low-rank assumption holds.

### 1.4 Notation

Bold lowercase letters are vectors and bold uppercase letters are matrices. We denote the set of integers $[n] := \{1, \ldots, n\}$. For vectors, $\|\cdot\|$ is the standard $\ell_2$ (Euclidean) norm. For matrices, $\|\cdot\|_{S_p}$ is the Schatten-$p$ norm, i.e., the $\ell_p$ norm of the vector of singular values, and $\|\cdot\|_{S_2} = \|\cdot\|_F$ is the Frobenius norm. The set of probability measures over $\mathbb{R}^d$ with finite 2nd moment is $\mathcal{P}_2(\mathbb{R}^d)$, and the subset of absolutely continuous measures is $\mathcal{P}_{2,ac}(\mathbb{R}^d)$.

### 1.5 Organization

First, in Section 2, we give the necessary background and outline our optimization program. Then, in Section 3, we provide our algorithmic framework for solving the Schatten OT problem. After this, Section 4 gives theoretical results about the structure of Schatten OT couplings. Finally, Section 5 presents experiments on synthetic and real data, highlighting the flexibility and advantages of our framework.

## 2 Background and Method

In this section, we first discuss background ideas in OT and compressed sensing, and then the Schatten OT method. We begin in Section 2.1 by describing discrete OT and its common regularizations. Then, Section 2.2 discusses background on Schatten-$p$ regularization in compressed sensing. After this, we define our Schatten-$p$ norm regularized OT, Schatten OT, in Section 2.3.

### 2.1 OT and Regularization

For simplicity of presentation, we focus on the discrete case; in Appendix A, we show how these ideas extend to the continuous setting. Consider two discrete measures $\mu = \sum_{i=1}^m a_i \delta_{\boldsymbol{x}_i}$ and $\nu = \sum_{j=1}^n b_j \delta_{\boldsymbol{y}_j}$, where $a_i, b_j \geq 0$ and $\sum_i a_i = \sum_j b_j = 1$. We let $\boldsymbol{X} \in \mathbb{R}^{d \times m}$ and $\boldsymbol{Y} \in \mathbb{R}^{d \times n}$ be matrices with the support points as columns. Without loss of generality, assume $n \geq m$. The transportation polytope $\mathcal{U}(\boldsymbol{a}, \boldsymbol{b})$ is the set of $m \times n$ nonnegative matrices whose rows sum to $\boldsymbol{a} = [a_1, \ldots, a_m]^\top$ and columns sum to $\boldsymbol{b} = [b_1, \ldots, b_n]^\top$. We also refer to these matrices as couplings between $\mu$ and $\nu$. In the transport problem, we are thinking of transporting $\mu$ to $\nu$. Therefore, we refer to $\mu$ as the *source distribution* and $\nu$ as the *target distribution*.

We assume an $m \times n$ matrix of costs $\boldsymbol{C}$, where $\boldsymbol{C}_{ij} = c(\boldsymbol{x}_i, \boldsymbol{y}_j)$, for some function $c : \mathbb{R}^d \times \mathbb{R}^d \to [0, \infty)$. The function $c$ is typically called the *ground cost*. As a concrete example, we can use the square of the Euclidean distance,

$$\boldsymbol{C}_{ij} = \|\boldsymbol{x}_i - \boldsymbol{y}_j\|^2. \tag{1}$$

OT seeks a minimum-cost coupling between the measures $\mu$ and $\nu$. It is formulated as a linear program over the transportation polytope,

$$\mathsf{OT}(\mu,\nu) = \min_{\boldsymbol{P}\in\mathcal{U}(\boldsymbol{a},\boldsymbol{b})}\langle\boldsymbol{P},\boldsymbol{C}\rangle. \tag{2}$$

When the cost corresponds to a power of a metric on some underlying space, the resulting OT cost can be used to define a metric on $\mathcal{P}_2(\mathbb{R}^d)$. For the rest of this paper, we will assume that $c(\boldsymbol{x}_i,\boldsymbol{y}_j) = \|\boldsymbol{x}_i - \boldsymbol{y}_j\|^2$. However, we note that our regularization can be applied to OT with any ground cost.

For a review of computational methods related to this linear program, see Peyré and Cuturi (2019). While there are many deep and interesting results related to OT, in practice, the direct use of OT in the form presented can suffer. In particular, OT suffers the curse of dimensionality; the worst-case statistical rate of estimation for the $p$-Wasserstein distance is $O(n^{-1/d})$, assuming that $\boldsymbol{x}_i$ and $\boldsymbol{y}_j$ are i.i.d. samples from some population measures. On the computational side, for large $n \asymp m$, the linear program incurs computational cost $O(n^3)$, and we must store an $O(n^2)$ variable in memory. To combat these issues, various regularizers have been considered, as we mentioned in the introduction. These regularized OT variants solve

$$\min_{\boldsymbol{P}\in\mathcal{U}(\boldsymbol{a},\boldsymbol{b})}\langle\boldsymbol{C},\boldsymbol{P}\rangle + \lambda R(\boldsymbol{P}),$$

where $R : \mathbb{R}^{m\times n} \to \mathbb{R}$ is the regularization function and $\lambda$ is a tunable parameter. An example is the entropy function $R(\boldsymbol{P}) = \sum_{ij}\boldsymbol{P}_{ij}(\log(\boldsymbol{P}_{ij}) - 1)$ (Cuturi, 2013).

In this paper, we propose a structured choice of regularizer based on the Schatten norms of affine transforms of $\boldsymbol{P}$, yielding a convex family of regularizers that can be swapped into any OT-based learning pipeline to encourage a low-dimensional structure.

## 2.2 Schatten-$p$ Regularization in Compressed Sensing

Schatten-$p$ regularization in compressed sensing served as a way to generalize $\ell_p$ regularization for sparse vector recovery. In particular, using Schatten-$p$ regularization is strictly more general than $\ell_p$ regularization because we can encode vectors as diagonal matrices, in which case $\|\boldsymbol{x}\|_p = \|\mathrm{diag}(\boldsymbol{x})\|_{S_p}$.

Perhaps the most popular regularization is Schatten-1 (nuclear norm) regularization. This is typically used to relax the rank of a matrix, and in a variety of settings, nuclear norm minimization has been shown to recover low-rank matrices (Recht et al., 2010; Candès and Tao, 2010; Candès et al., 2011).

These methods have been applied in a variety of settings. Some applications of these methods have included matrix completion and recommender systems (Nie et al., 2012), multitask learning (Zhang et al., 2018), high-dimensional covariance estimation (Gavish and Donoho, 2017), and image processing (Xie et al., 2016).

Optimization with nuclear norms, or more generally Schatten-$p$ norms, involves a variety of algorithms. For example, nuclear norm minimization can involve saddle point or proximal algorithms (Nesterov and Nemirovski, 2013), the latter of which involves singular value thresholding (Cai et al., 2010). Other algorithmic paradigms include primal-dual methods (Chambolle and Pock, 2011) or ADMM (Yuan and Yang, 2013). To solve the more general case of Schatten-$p$ regularized problems, for $0 < p \le 2$, one typically resorts to Lagrangian-style methods (Nie et al., 2012) or iteratively reweighted nuclear norm-style methods (Lu et al., 2015). Another popular approach to nuclear norm minimization problems involves factor splitting to avoid SVD computations (Srebro et al., 2004; Fan et al., 2019).

## 2.3 Discrete Schatten-$p$ Regularized OT

We now come to the main innovation of our work. We study a new variant of regularized OT problems using Schatten-$p$ norms. We define the Schatten OT problem as

$$\mathsf{Sch\text{-}OT}(\mu,\nu;\{(\lambda_i,p_i,q_i,\mathcal{A}_i)\}) := \min_{\boldsymbol{P}\in\mathcal{U}(\boldsymbol{a},\boldsymbol{b})}\langle\boldsymbol{C},\boldsymbol{P}\rangle + \sum_i \lambda_i\|\mathcal{A}_i(\boldsymbol{P})\|_{S_{p_i}}^{q_i}. \tag{3}$$

The idea of this program is to regularize toward simpler couplings with respect to the maps $\mathcal{A}_i$. Notice that the Schatten OT problem relies on three sets of parameters: the regularization strengths $\lambda_i \ge 0$, the

Schatten powers and exponents $p_i, q_i > 0$, and maps $\mathcal{A}_i : \mathcal{U}(\boldsymbol{a}, \boldsymbol{b}) \to \mathbb{R}^{k_i \times l_i}$. Provided that $p_i, q_i \geq 1$ and $\mathcal{A}_i$ are affine, it is easily seen that Schatten OT is a convex program. With only one regularization term, this simplifies to $\mathsf{Sch\text{-}OT}(\mu, \nu; (\lambda, p, q, \mathcal{A})) = \min_{\boldsymbol{P} \in \mathcal{U}(\boldsymbol{a}, \boldsymbol{b})} \langle \boldsymbol{C}, \boldsymbol{P} \rangle + \lambda \|\mathcal{A}(\boldsymbol{P})\|_{S_p}^q$.

We believe that convexity and generality are primary benefits of Schatten OT. It is general enough to cover many existing regularizations in the literature, as we will demonstrate shortly. The convexity of the problem leads to solutions that are easy to characterize, as we will show in our theory section. It also enables efficient solvers using convex optimization techniques.

While many past regularizations for OT fall into this framework, some do not. In particular, classical entropic regularization cannot be recovered from (3), because the negative entropy is not a Schatten norm of an affine function of $\boldsymbol{P}$. Nevertheless, entropy can be combined with Schatten OT in a straightforward way by considering objectives with both the Schatten OT and entropic regularizers. For convex Schatten regularization, this problem remains convex and can be solved by the same KL mirror descent algorithm: the entropic term contributes an additional term to the gradient, while the KL projection onto the transport polytope is still implemented via Sinkhorn scaling. In this way, we can interpolate between standard entropic OT and the purely Schatten-regularized regime studied in this work. Also, in Appendix A, we illustrate how to extend Schatten OT to the continuous setting.

**Low-rank and Sparse Couplings:** As a first example, consider the affine map $\mathcal{A}(\boldsymbol{P}) = \boldsymbol{P}$. Then, depending on the choice of $p$, Schatten OT encourages low-rank or sparse couplings. In particular, choosing $q = p = 1$ promotes low-rank solutions while preserving convexity, while choosing $p < 1$ encourages low-rank solutions but is non-convex. We do not study this nonconvex case here. This yields a principled, optimization-based analog to the low-rank factorization pursued by works such as Forrow et al. (2019); Scetbon et al. (2021). On the other hand, $q = p = 2$ corresponds to the case of quadratically regularized OT (Blondel et al., 2018; Lorenz et al., 2021), since the Schatten-2 norm is just the Frobenius norm. Perhaps surprisingly, quadratic regularization in optimal transport tends to encourage sparse solutions (González-Sanz and Nutz, 2024). Group sparsity (Blondel et al., 2018) can be achieved through proper choice of the affine maps in (3) and setting $p_i = 2$, $q_i = 1$.

**Elastic costs:** We can also recover some of the elastic cost regularizations of Cuturi et al. (2023); Klein et al. (2024); Chen et al. (2025). In particular, we can take $q = p = 1$ and let $\mathcal{A}(\boldsymbol{P})$ be the affine map

$$\boldsymbol{P} \mapsto \mathrm{diag}\Big( (\boldsymbol{P}_{ij}(\boldsymbol{x}_i - \boldsymbol{y}_j))_{i=1\ldots m, j=1\ldots n} \Big).$$

Then, the Schatten OT penalty corresponds to the $\ell_1$ elastic cost. Group-sparse elastic costs can be recovered from sums of Schatten regularizations. We can also recover the subspace elastic costs of Klein et al. (2024) by taking $q = p = 2$ and the affine map

$$\boldsymbol{P} \mapsto \mathrm{diag}\Big( (\boldsymbol{Q}_L \boldsymbol{P}_{ij}(\boldsymbol{x}_i - \boldsymbol{y}_j))_{i=1\ldots m, j=1\ldots n} \Big),$$

where $\boldsymbol{Q}_L$ is the projection onto the orthogonal complement of $L$. Note that, analogously to Klein et al. (2024), one could include an additional minimization over $L$ in the Schatten OT formulation. This then defines a family of learnable Schatten OT problems. We discuss this possibility further in the appendix.

**Barycentric projection maps and displacements:** The formulation can be used to penalize map estimators directly. In particular, given a transport plan $\boldsymbol{P}$, one can estimate a transport map using the *barycentric projection*

$$T_{\boldsymbol{P}}(\boldsymbol{x}_i) = \frac{1}{a_i} \sum_{j=1}^{n} \boldsymbol{P}_{ij} \boldsymbol{y}_j = \frac{1}{\boldsymbol{a}_i} (\boldsymbol{Y} \boldsymbol{P}^\top)_{:,i}.$$

Notice that this map is linear in $\boldsymbol{P}$. Thus, we can penalize the barycentric projection map in our program by letting $\mathcal{A}(\boldsymbol{P}) = T_{\boldsymbol{P}}(\boldsymbol{X})\boldsymbol{A}^{-1/2}$, where we define $\boldsymbol{A} = \mathrm{diag}(\boldsymbol{a})$. As a further example, we could encourage displacements to be low-rank rather than the map itself. We call $T_{\boldsymbol{P}}(\boldsymbol{x}_i) - \boldsymbol{x}_i$ the *barycentric displacement*. Then, to encourage these to be simple, we could use $\mathcal{A}(\boldsymbol{P}) = \boldsymbol{Y} \boldsymbol{P}^\top \boldsymbol{A}^{-1/2} - \boldsymbol{X} \boldsymbol{A}^{1/2}$. In both cases, the additional scaling of $\boldsymbol{A}^{1/2}$ allows the population limit of the program to be well defined.

**Covariance regularization:** All of the maps discussed so far include zeroth and first moments of the support points $(\boldsymbol{x}_i, \boldsymbol{y}_j)$. However, our formulation is flexible enough to include higher moments of our data distribution. For example, we can take $\mathcal{A}$ to be an affine function of the covariance induced by $\boldsymbol{P}$,

$$\boldsymbol{\Sigma_P} = \sum_{ij} \boldsymbol{P}_{ij} \begin{pmatrix} \boldsymbol{x}_i \\ \boldsymbol{y}_j \end{pmatrix} \begin{pmatrix} \boldsymbol{x}_i \\ \boldsymbol{y}_j \end{pmatrix}^\top,$$

which is linear in $\boldsymbol{P}$.

We could penalize the Schatten-1 norm of the cross-covariance $\sum_{ij} \boldsymbol{P}_{ij} \boldsymbol{x}_i \boldsymbol{y}_j^\top$, which is an affine function of $\boldsymbol{\Sigma_P}$. If the vectors $\boldsymbol{x}_i$ and $\boldsymbol{y}_j$ are whitened (i.e., they each have identity covariance), then the singular values of this matrix correspond to the canonical correlations. Minimizing the Schatten-1 norm in this case corresponds to minimizing the sum of canonical correlations, which seeks to increase independence between $X$ and $Y$. Note that if we take the $\mathcal{A}(\boldsymbol{P}) = \sum_{ij} \boldsymbol{P}_{ij}(\boldsymbol{x}_i - \boldsymbol{y}_j)(\boldsymbol{x}_i - \boldsymbol{y}_j)^\top$ and $p = 1$, then the regularization is just the quadratic OT cost, $\|\sum_{ij} \boldsymbol{P}_{ij}(\boldsymbol{x}_i - \boldsymbol{y}_j)(\boldsymbol{x}_i - \boldsymbol{y}_j)^\top\|_{S_1} = \sum_{ij} \boldsymbol{P}_{ij}\|\boldsymbol{x}_i - \boldsymbol{y}_j\|^2$.

These illustrate just a few of the potential choices for adding covariance regularization to OT. In the appendix, we discuss covariance regularization in the context of Schatten OT for Gaussians. An in-depth study of these is left to future work.

## 3 A Mirror Descent Algorithm

The Schatten OT program in (3) is convex whenever $p_i, q_i \geq 1$ and $\mathcal{A}_i$ are affine, but solving it directly with off-the-shelf convex solvers (e.g., CVXPY (Diamond and Boyd, 2016) or interior point methods) is only feasible for small problems, as the transportation polytope $\mathcal{U}(\boldsymbol{a}, \boldsymbol{b})$ involves $\mathcal{O}(nm)$ variables and constraints. To address large-scale settings, we turn to first-order optimization methods. A particularly effective choice for optimization over the transport polytope is mirror descent with Kullback–Leibler (KL) geometry (Kemertas et al., 2025). We use this algorithm for its simplicity and leave the analysis of more general methods, such as primal-dual algorithms or ADMM, to future work.

### 3.1 Mirror Descent for Schatten OT

Following the approach of Kemertas et al. (2025), we develop mirror descent using the KL geometry on the transport polytope. This choice is natural, since using the KL geometry replaces a costly Euclidean projection with efficient Sinkhorn scaling. A few iterations of this method, followed by rounding, are effective at projecting to the polytope (Altschuler et al., 2017).

Assuming that $\mathcal{A}(\boldsymbol{P}) \neq 0$, we can compute a subgradient of the Schatten-$p$ norm term in the Schatten OT problem as

$$q\|\mathcal{A}(\boldsymbol{P})\|_{S_p}^{q-p} \mathcal{A}^\star \left(\boldsymbol{U}\boldsymbol{\Sigma}^{p-1}\boldsymbol{V}^\top\right) \in \partial\|\mathcal{A}(\boldsymbol{P})\|_{S_p}^q,$$

where $\partial$ denotes the subdifferential. This provides a computable subgradient of $F(\boldsymbol{P})$ at each iteration, provided that we can compute a singular value decomposition. For $p, q > 1$, this is a bona fide gradient.

The mirror descent iteration involves the following steps. Define the cost function we wish to minimize as

$$F(\boldsymbol{P}) := \langle \boldsymbol{C}, \boldsymbol{P} \rangle + \lambda\|\mathcal{A}(\boldsymbol{P})\|_{S_p}^q.$$

Then, given an initial coupling $\boldsymbol{P}^0$ and sequence of step sizes $(\tau^k)_{k=0}^\infty$, we iterate the following steps:

1. Form the SVD $\mathcal{A}(\boldsymbol{P}^k) = \boldsymbol{U}^k \boldsymbol{\Sigma}^k \boldsymbol{V}^{k\top}$ and the subgradient

$$\boldsymbol{G}^k = \boldsymbol{C} + \lambda \cdot q\|\mathcal{A}(\boldsymbol{P}^k)\|_{S_p}^{q-p} \mathcal{A}^\star \left(\boldsymbol{U}^k(\boldsymbol{\Sigma}^k)^{p-1}\boldsymbol{V}^{k\top}\right).$$

2. Use multiplicative-weights update

$$\widehat{\boldsymbol{P}}_{ij}^{k+1} \propto \boldsymbol{P}_{ij}^k \exp(-\tau^k \boldsymbol{G}_{ij}^k).$$

3. Project back to the transport polytope

$$\boldsymbol{P}^{k+1} = \Pi^{\mathrm{KL}}_{\mathcal{U}(\boldsymbol{a},\boldsymbol{b})}(\widehat{\boldsymbol{P}}^{k+1}_{ij}).$$

Here, $\Pi^{\mathrm{KL}}_{\mathcal{U}(\boldsymbol{a},\boldsymbol{b})}$ is the projection onto the transport polytope with respect to the KL divergence, which can be implemented via Sinkhorn scaling. Thus we see that each iteration requires one SVD on $\mathcal{A}(\boldsymbol{P}^k)$ (to evaluate the Schatten subgradient) and one Sinkhorn projection.

By standard mirror descent theory (Beck and Teboulle, 2003; Nemirovsky and Yudin, 1983; Bubeck, 2015), the method achieves an $O(1/\sqrt{T})$ convergence rate for convex objectives like $F$ when $p, q \geq 1$. The KL geometry ensures that nonnegativity is automatically preserved, and averaging can be used to guarantee convergence of the objective values. While more sophisticated primal-dual or ADMM variants can be competitive in some regimes, carefully tuning and analyzing them for the full generality of (3) is nontrivial, and we leave a systematic study to future work.

The most expensive parts of the iteration are the SVD computation and the Sinkhorn projection. It is possible to use an adaptive low-rank approximation of $\mathcal{A}(\boldsymbol{P}^k)$ throughout the iterations to increase computational efficiency. It would also be interesting to attempt to use sketching methods to approximate low-rank solutions to this problem (Yurtsever et al., 2021).

The choice of step size is essential. In general, since the problem is convex but not smooth in general, one could take $\tau^k \propto 1/\sqrt{k+1}$, which yields the $O(1/\sqrt{T})$ convergence rate in objective value. In our experiments, we can observe faster convergence in specific settings. For example, when $p = q = 1$ and $\mathcal{A}$ has simple structure ($\mathcal{A}(\boldsymbol{P}) = \boldsymbol{P}$ or $\mathcal{A}(\boldsymbol{P}) = \boldsymbol{Y}\boldsymbol{P}^\top\boldsymbol{A}^{-1/2}$), mirror descent can obtain faster convergence with a geometrically diminishing step size, the same schedule used in past work with sharp minima (Davis et al., 2018; Maunu et al., 2019).

In practice, we must tune the regularization parameters $\lambda$ and $p$. For $\lambda$, we can start from the unregularized OT cost and increase $\lambda$ until the effective rank, $\|\mathcal{A}(\boldsymbol{P})\|_{S_1}/\|\mathcal{A}(\boldsymbol{P})\|_{S_\infty}$, drops while the cost increase is modest (see the examples in Figures 1 and 2). On the other hand, the choice of $p$ is determined by what type of inductive bias one wants to enforce on $\mathcal{A}(\boldsymbol{P})$. For $p = 1$, we see the sharpest rank reduction; $p = 2$ softens shrinkage but can enforce sparsity; $p = \infty$ amounts to operator-norm penalization.

## 3.2 Alternative Algorithms

A core computational bottleneck in large-scale OT solvers is the repeated enforcement of the row/column constraints, i.e., projecting onto the transportation polytope $\mathcal{U}(\boldsymbol{a}, \boldsymbol{b})$. We adopted mirror descent with Kullback-Leibler (KL) geometry in this work because projection onto $\mathcal{U}(\boldsymbol{a}, \boldsymbol{b})$ coincides with Sinkhorn scaling (Sinkhorn, 1967; Cuturi, 2013). One could use gradient descent with respect to another geometry, such as the Euclidean geometry. In this case, the projection corresponds to a constrained least-squares projection, which is substantially more expensive to compute. For the Sinkhorn method, there exist highly optimized implementations and near-linear-time approximation guarantees (Altschuler et al., 2017). Using KL mirror descent, therefore, lets us reuse the same projection primitive that underlies scalable entropic OT solvers, while optimizing a much more general convex objective.

There are several natural alternatives, including primal-dual splitting methods (Chambolle and Pock, 2011) and ADMM (Yuan and Yang, 2013). These approaches can be attractive when the problem admits a convenient splitting (e.g., separating the transport constraints from a proximal step for the Schatten penalty), but they typically introduce auxiliary variables and dual constraints. In our setting, this can obscure the transport-polytope projection structure and make it harder to directly reuse Sinkhorn-based OT primitives.

Regarding SVD computations, the cost is highly formulation-dependent within the Schatten OT framework. In several important cases, the Schatten term admits simple gradients/subgradients that avoid a full SVD altogether; for example, when $p = q = 2$ and $\mathcal{A}(\boldsymbol{P}) = \boldsymbol{P}$, the penalty reduces to a Frobenius norm, and similarly for group-sparse Frobenius penalties on blocks of $\boldsymbol{P}$. Likewise, for elastic-cost-style constructions in which $\mathcal{A}(\boldsymbol{P})$ is diagonal or block-diagonal, the required spectral computations reduce to elementwise or small-block operations. Even in the nuclear-norm setting ($p = 1$), when solutions or iterates are numerically

low-rank, one can use thin/truncated SVDs or randomized approximations to reduce per-iteration cost. Exploring primal-dual variants that systematically exploit such structure is an interesting direction for future work.

# 4 Theory

In this section, we present our main theoretical results on the Schatten OT program. First, Section 4.1 uses convex optimization theory to outline the structure of solutions to the Schatten OT problem. After this, Section 4.2 uses this structure to prove two theorems that demonstrate Schatten OT's ability to recover low-rank couplings and barycentric displacements. Finally, in Section 4.3, we finish with a discussion of our theoretical results.

## 4.1 General Structural Theorems

Let $\boldsymbol{P}^\star$ be an optimal solution of the optimization problem (3) with $p, q \geq 1$. Since this is a constrained convex optimization problem in $\boldsymbol{P}$, we can appeal to standard theory. The solution is characterized by the KKT conditions, which state that there exists a subgradient $\boldsymbol{G}^\star \in \partial \|\mathcal{A}(\boldsymbol{P}^\star)\|_{S_p}^q$ and dual variables $\boldsymbol{u}^\star \in \mathbb{R}^m$ and $\boldsymbol{v}^\star \in \mathbb{R}^n$ such that

$$\boldsymbol{C} + \lambda \boldsymbol{G}^\star + \boldsymbol{u}^\star \mathbf{1}_n^\top + \mathbf{1}_m \boldsymbol{v}^{\star\top} = \mathbf{0},$$
$$\boldsymbol{P}^\star \geq 0, \quad \boldsymbol{P}^\star \mathbf{1} = \boldsymbol{a}, \quad \boldsymbol{P}^{\star\top} \mathbf{1} = \boldsymbol{b}.$$

Comparing these conditions to the optimality conditions for standard OT, we notice that the only difference is the inclusion of the $\lambda \boldsymbol{G}^\star$ in the first-order stationarity condition. Therefore, the optimality conditions for Schatten OT are precisely those for an OT problem with the *tilted cost*

$$\boldsymbol{S}(\lambda, \boldsymbol{G}^\star) := \boldsymbol{C} + \lambda \boldsymbol{G}^\star \in \mathbb{R}^{m \times n}. \tag{4}$$

where $\boldsymbol{G}^\star$ is some subgradient of $\|\mathcal{A}(\cdot)\|_{S_p}^q$ at $\boldsymbol{P}^\star$. The obstacle to our directly applying this result is that we do not know $\boldsymbol{G}^\star$, since that would require knowing $\boldsymbol{P}^\star$.

We can state this characterization as the following proposition.

**Proposition 1.** *The coupling $\boldsymbol{P}^\star$ is optimal for* (3) *if and only if there exists a subgradient $\boldsymbol{G}^\star$ of $\|\mathcal{A}(\boldsymbol{P}^\star)\|_{S_p}^q$ such that*

$$\boldsymbol{P}^\star \in \mathrm{argmin}_{\boldsymbol{P} \in \mathcal{U}(\boldsymbol{a}, \boldsymbol{b})} \langle \boldsymbol{S}(\lambda, \boldsymbol{G}^\star), \boldsymbol{P} \rangle.$$

While we cannot apply this to the direct computation of $\boldsymbol{P}^\star$, we can use it to characterize solutions to the Schatten OT problem. In the following section, we develop this idea to prove the recovery of low-rank structure in the Schatten OT problem.

## 4.2 Discrete Recovery Theorems

In this section, we prove low-rank recovery theorems for Schatten OT. While these are restrictive toy examples, they represent the first such exact recovery results in the OT literature. We believe this is a first step toward applying compressed sensing ideas to regularized OT problems. It is an open question for future work to extend these ideas to the recovery of simple couplings in more complex settings. We begin in Section 4.2.1 with a recovery result for low-rank couplings. Then, Section 4.2.2 presents a consequence on the recovery of a low-rank set of barycentric displacements. These recovery guarantees play the same role as recovery results in compressed and matrix sensing: they certify when the imposed structural bias is correct and will be recovered, enabling principled interpretability.

### 4.2.1 Low-Rank Coupling Recovery

We now assume that both the source and the target consist of $R$ well-separated clusters, each with the same cardinality. We show that, for a nontrivial interval of regularization strengths $\lambda$, the nuclear-norm penalized

OT problem recovers a rank-$R$, block-diagonal coupling that matches each source cluster uniformly to its corresponding target cluster. We assume uniform marginals $a_i = 1/m, b_j = 1/n$ for all $i, j$.

Our first assumption is on the clustered structure of $\mu$ and $\nu$.

**Assumption 2.** *For two measures $\mu = \sum_{i=1}^{m} a_i \delta_{\boldsymbol{x}_i}$ and $\nu = \sum_{j=1}^{n} b_j \delta_{\boldsymbol{y}_j}$, $n = Rg$, $m = Rg$ for integers $R, g \geq 1$. The source indices $[m]$ and target indices $[n]$ are partitioned into clusters $S_1, \ldots, S_R$ and $T_1, \ldots, T_R$, respectively, where $|S_t| = |T_t| = g$ for all $t$.*

Let $B(\boldsymbol{z}, \rho)$ denote the closed Euclidean ball of radius $\rho > 0$ around $\boldsymbol{z} \in \mathbb{R}^d$. Our goal is to construct a setting where the cluster $S_t$ is uniformly matched to $T_t$. We make the following assumptions about the locations of the source and target points.

**Assumption 3.** *For two measures $\mu = \sum_{i=1}^{m} a_i \delta_{\boldsymbol{x}_i}$ and $\nu = \sum_{j=1}^{n} b_j \delta_{\boldsymbol{y}_j}$,*

1. *The source points lie in disjoint balls, $\boldsymbol{x}_i \in B(\boldsymbol{c}_t, \rho)$ for $i \in S_t$, and the target points lie in disjoint balls, $\boldsymbol{y}_j \in B(\boldsymbol{d}_t, \rho)$ for $j \in T_t$.*

2. *Within matched clusters $S_t$ and $T_t$, $\|\boldsymbol{x}_i - \boldsymbol{y}_j\| = \|\boldsymbol{x}_i - \boldsymbol{y}_{j'}\|$ for all $i \in S_t$ and $j, j' \in T_t$ for $t = 1, \ldots, R$.*

3. *The minimum inter-cluster distance $\Gamma := \min_{s \neq t} \|\boldsymbol{c}_t - \boldsymbol{d}_s\|$ and the maximum intra-cluster distance $\gamma := \max_t \|\boldsymbol{c}_t - \boldsymbol{d}_t\|$ satisfy*

$$\Gamma > \gamma + 4\rho > 0. \tag{5}$$

Notice that, under our separation condition (5), the OT coupling actually respects the cluster structure, in the sense that it must match points in $S_t$ to $T_t$. Furthermore, any plan that matches $\boldsymbol{x}_i$ to $\boldsymbol{y}_j$ within clusters (when $i \in S_t$, $j \in T_t$) is optimal. However, these matched clusterings are not low-rank; they are full-rank. On the other hand, as we will show in the following theorem, a low-rank matching can be recovered from Schatten OT.

**Theorem 4.** *Let Assumptions 2 and 3 hold, $\boldsymbol{C}$ be the quadratic cost, and let the excess cost for an across-cluster matching be*

$$\Delta_{\min} := \min_{\substack{s \neq t \in [R] \\ i \in S_t, j \in T_s, j' \in T_t}} \left\{ \|\boldsymbol{x}_i - \boldsymbol{y}_j\|_2^2 - \|\boldsymbol{x}_i - \boldsymbol{y}_{j'}\|_2^2 \right\}.$$

*Then, for any regularization parameter $\lambda$ satisfying*

$$0 \leq \lambda < g \cdot \Delta_{\min}, \tag{6}$$

*the minimizer of $\langle \boldsymbol{C}, \boldsymbol{P} \rangle + \lambda \|\boldsymbol{P}\|_{S_1}$ is a rank $R$ coupling supported blockwise on $\bigcup_{t=1}^{R} (S_t \times T_t)$ that is uniform within clusters.*

The essential idea of the proof is to ensure that 1) $\boldsymbol{P}^\star$ is the unique coupling that respects the cluster structure and also minimizes the nuclear norm, and 2) there is no way to make the nuclear norm term even smaller by using across-cluster matches without incurring more cost.

### 4.2.2 Low-Rank Displacement Recovery when $p = 1$

We now give a concrete example of how to recover a coupling with low-rank displacements. For our affine map, we consider the weighted barycentric displacement matrix $\mathcal{A}(\boldsymbol{P})$,

$$\mathcal{A}(\boldsymbol{P}) := \left( T_{\boldsymbol{P}}(\boldsymbol{X}) - \boldsymbol{X} \right) \boldsymbol{A}^{1/2} = \boldsymbol{Y} \boldsymbol{P}^\top \boldsymbol{A}^{-1/2} - \boldsymbol{X} \boldsymbol{A}^{1/2}.$$

We also again assume that $p = q = 1$ in the Schatten OT formulation. For our recovery result, we make the following assumptions.

**Assumption 5** (Symmetric two-target clusters with separation). *Fix orthonormal vectors $\boldsymbol{u}, \boldsymbol{v} \in \mathbb{R}^d$ and an integer $R \geq 2$. Let $0 < \mu_1 < \cdots < \mu_R$ be distinct scalars and put $\boldsymbol{m}_t := \mu_t \boldsymbol{u} \in \mathbb{R}^d$ for $t \in [R]$. Suppose:*

1. *The source set $[m]$ is partitioned into nonempty clusters $S_1, \ldots, S_R$ and there exists $\rho > 0$ such that $\boldsymbol{x}_i \in \boldsymbol{m}_t + [-\rho, \rho]\boldsymbol{u}$ for all $i \in S_t$.*

2. *For some $\varepsilon > 0$, the target support consists of the $2R$ points*

$$\boldsymbol{y}_{t,+} = \boldsymbol{m}_t + \varepsilon\boldsymbol{v}, \ \ \boldsymbol{y}_{t,-} = \boldsymbol{m}_t - \varepsilon\boldsymbol{v}, t \in [R],$$

   *with target masses $b_{t,+} = b_{t,-} = \frac{1}{2}\sum_{i \in S_t} a_i$.*

3. *The minimal separation between clusters is lower bounded*

$$\Lambda := \min_{s \neq t} |\mu_t - \mu_s| > 2\rho.$$

By symmetry, it is easy to see that all couplings that assign the source points in cluster $S_t$ to $\boldsymbol{y}_{t,+}$ or $\boldsymbol{y}_{t,-}$ are optimal. In particular, it does not matter how the points are assigned within clusters. For $i \in S_t$ and $s \neq t$, it is also convenient to define the inter-cluster cost gap

$$\Delta_{i,s} := \|\boldsymbol{x}_i - \boldsymbol{y}_{s,\pm}\|_2^2 - \|\boldsymbol{x}_i - \boldsymbol{y}_{t,\pm}\|_2^2 = (\mu_t - \mu_s)^2 + 2(\mu_t - \mu_s)\xi_i, \tag{7}$$

where $\boldsymbol{x}_i = \boldsymbol{m}_t + \xi_i\boldsymbol{u}$ and $|\xi_i| \leq \rho$.

We now state the main recovery theorem for this setting. It says that, under our assumption, we can exactly recover a coupling with a rank-1 barycentric displacement.

**Theorem 6.** *Let Assumption 5 hold and suppose $\boldsymbol{C}$ is the quadratic cost. Define the explicit threshold*

$$\lambda_{\max} := \Lambda - 2\rho > 0.$$

*Then for every $\lambda \in [0, \lambda_{\max})$, the unique minimizer of $\langle \boldsymbol{C}, \boldsymbol{P}\rangle + \lambda\|\mathcal{A}(\boldsymbol{P})\|_{S_1}$, where $\mathcal{A}(\boldsymbol{P})$ is the weighted barycentric displacement, matches $\boldsymbol{x}_i$ to $\{\boldsymbol{y}_{t,\pm}\}$ for $i \in S_t$. Furthermore, it yields a rank-1 barycentric displacement.*

### 4.3 Discussion

While we demonstrate low-rank recovery only in toy examples here, our methodology highlights the advantages of using convex formulations. In particular, it is easier to verify that the recovered solution is low-rank. In fact, these are the first guarantees of low-rank recovery within an OT problem in the literature. This contrasts with nonconvex methods, which currently lack guarantees (Forrow et al., 2019; Klein et al., 2024).

In the future, it would be interesting to demonstrate exact recovery in more general settings using Schatten-$p$ regularization when $p < 1$. It would also be interesting to develop more general recovery conditions to move beyond the toy examples considered here.

## 5 Experiments

In this section, we give some simulations on real data that demonstrate the advantages of the Schatten OT formulation. First, in Section 5.1, we demonstrate Schatten OT's ability to recover low-rank couplings and barycentric projection maps. Then, in Section 5.2, we examine the convergence rate of mirror descent to solve the Schatten OT problem. Finally, Section 5.3 gives an experiment on real data that demonstrates the ability of Schatten OT to recover simpler couplings with 4i perturbation data.

In these experiments, we also compare against other baseline regularized optimal transport methods. For recovery of low-rank couplings and low-rank barycentric projections, we compare against the Sinkhorn algorithm (Cuturi, 2013) as well as Low-Rank Sinkhorn Factorization (Scetbon et al., 2021). Both methods are run using the Python Optimal Transport package.[1] For the recovery of low-rank displacements, we compare against Sinkhorn and entropically regularized Subspace Elastic OT (Klein et al., 2024). Since publicly available code for Subspace Elastic OT was not available, we implement this as alternating minimization over couplings and subspaces, where the coupling is found via entropic optimal transport with a fixed regularization parameter and the subspace is found via PCA.

---

[1]https://pythonot.github.io/

### 5.1 Low-rank Recovery

To begin, we examine properties of the Schatten OT problem, and we use CVXPY to solve the convex program exactly.

In our first experiment, we examine the Schatten OT's ability to recover low-rank couplings. To measure the quality of the recovered $\boldsymbol{P}^\star$, we use two metrics: effective rank and transport cost. The latter is defined as $\langle \boldsymbol{C}, \boldsymbol{P}^\star \rangle$. The former is the ratio of the nuclear norm to the operator norm:

$$\text{Effective Rank}(\boldsymbol{B}) = \frac{\|\boldsymbol{B}\|_{S_1}}{\|\boldsymbol{B}\|_{S_\infty}}. \tag{8}$$

In our experiments, the support of $\mu$ consists of two clusters centered at $(-2, 2)$ and $(-2, -2)$, and $\nu$ consists of two clusters centered at $(2, 2)$ and $(2, -2)$. The points within each cluster are Gaussian. We sample 10 points from each cluster, so that $n = m = 20$. The results are averaged over five randomly generated datasets.

In the first experiment displayed in the top row of Figure 1, we use a variance of 0.04 for each Gaussian component. On the left, we show the effective rank versus regularization strength $\lambda$ for $p = 1, 2$, and $\infty$. On the right, we display the excess transport cost compared to standard optimal transport with the quadratic cost. As we can see, nuclear norm regularization can significantly reduce the effective rank without substantially increasing the transport cost. Using the Schatten-2 norm can also reduce the effective rank, albeit more gradually. This contrasts with Sinkhorn, which recovers lower-rank couplings since it is biased toward product couplings but incurs a higher transport cost. Low-rank Sinkhorn recovers a good low-rank coupling for well-chosen ranks and a small regularization, but if the regularization parameter is too large, it becomes unstable.

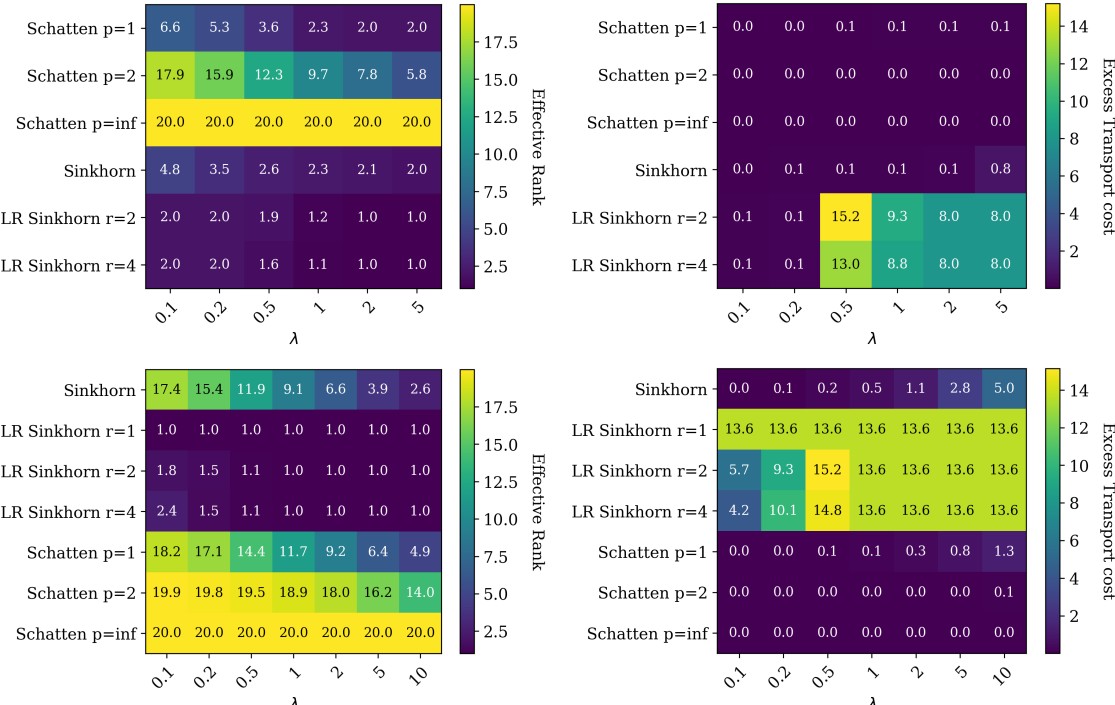

Figure 1: Solution quality of Schatten OT versus regularization parameter for Gaussian mixture data. Top row: small variance. Bottom row: large variance. On the left, we show the effective rank of the found coupling for each method; on the right, we display its excess transport cost. As we can see, Schatten-1 regularization can greatly simplify the transport plan without substantially increasing transport costs.

In the bottom row of Figure 1, we repeat the experiment from the top row with one modification: we use a larger within-cluster variance of 2. As we can see, it is more challenging to find a low-rank transport plan, and when one is found, it increases the transport cost more substantially.

The next experiment simulates the recovery of a low-rank transport displacement. Here, we wish to estimate a transport from a measure $\mu$ to a measure $\nu$. We assume that $\mu$ is standard Gaussian, and samples from $\nu$ are generated by $\boldsymbol{y}_i = \boldsymbol{x}_i + \xi_i \boldsymbol{u}$, where $\boldsymbol{u}$ is a random unit vector and $\xi_i$ is standard Gaussian. We now wish to recover a coupling with a low-rank barycentric map, and repeat the same experiment as in Figure 1 but now use $\mathcal{A}(\boldsymbol{P}) = \boldsymbol{Y}\boldsymbol{P}^\top \boldsymbol{A}^{-1/2} - \boldsymbol{X}\boldsymbol{A}^{1/2}$. Figure 2 displays the results of this experiment. As we can see, Schatten-1 regularization again recovers a coupling with lower-rank displacements without substantially increasing transport cost. Here, Sinkhorn does not recover low-rank displacements. Subspace elastic OT performs well, but we note that one needs to specify the rank, while Schatten OT automatically detects low-rank structure.

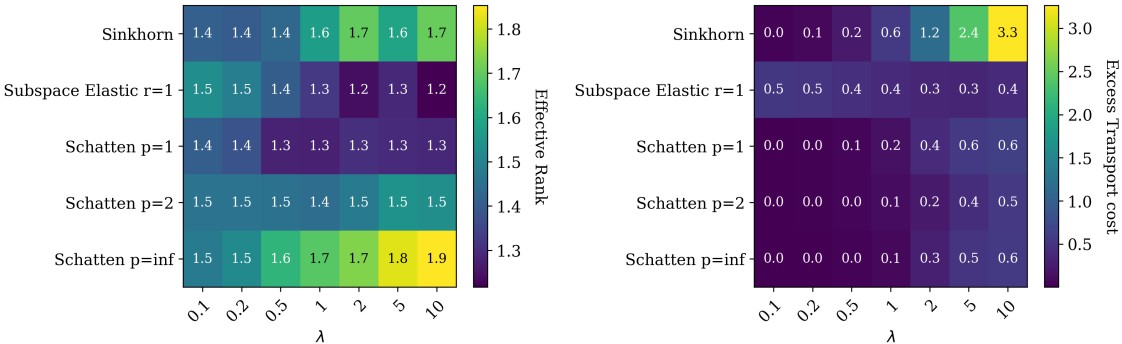

Figure 2: Solution quality of Schatten OT versus regularization parameter for Gaussian data with a low-rank perturbation. In the left display, we show the effective rank of the found barycentric displacements, and in the right display, we show the excess transport cost of the found coupling. As we can see, Schatten-1 regularization again simplifies the transport plan, but the transport cost increases substantially across the board. Subspace elastic OT performs well but requires one to know the rank in advance as an extra parameter.

## 5.2 Convergence Rates

Next, we examine the convergence rate of the mirror descent algorithm with Schatten-1 regularization in two settings. In the first setting, we show sublinear convergence; in the second, linear convergence.

In the left display of Figure 3, we use a setting where we do not expect a low-rank coupling to be easy to find. We set $\lambda = 0.1$, and the data $\mu$ is a mixture of Gaussians with centers at $(-2, \pm 2)$ and $\nu$ is a mixture of Gaussians with centers at $(2, \pm 2)$. The variance is set to be 1, and $n = m = 20$. We observe slow sublinear convergence. Note the log scale on the x-axis.

In the right display of Figure 3, we use the same setup as before, except now we set $\lambda = 10$ and let the clusters have variance 0.04. Now, since the recovered coupling is low-rank, mirror descent with a geometrically diminishing step size converges linearly. This implies that the objective is sufficiently sharp, which can be exploited by this step-size schedule.

## 5.3 4i Perturbation Example

Our final experiments are on transport estimation in a biological example. We consider a standard setting in single-cell perturbation analysis, where each experimental condition is represented by an empirical distribution in a high-dimensional expression space. A central goal is to learn how the cellular state distribution changes under intervention, i.e., to model the distributional shift between conditions. In this context, an optimal transport coupling between the two empirical measures can be viewed as a data-driven alignment

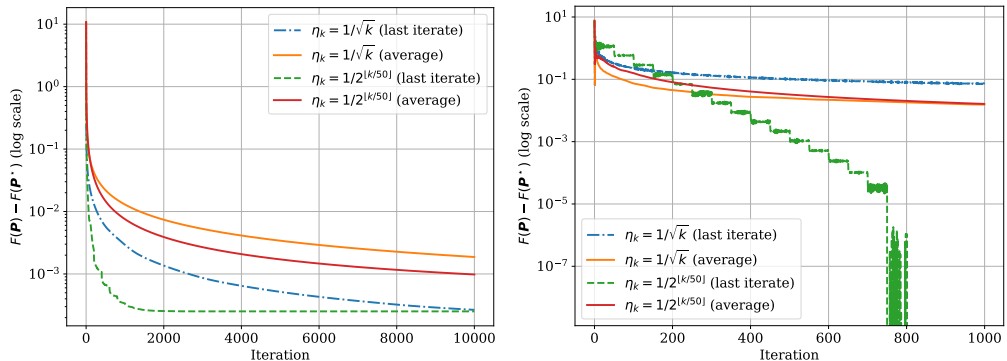

Figure 3: Plot of log excess cost versus iteration for the mirror descent algorithm on the Schatten OT problem. Left: In this experiment, the regularization parameter is small, and the variance of the Gaussian mixture components is large. This shows sublinear convergence of the algorithm, as is expected by the theory. Right: we reduce the variance and increase the regularization parameter, resulting in an optimal low-rank coupling. Here, we see that the geometrically diminishing step size converges linearly.

between populations of cells, and the associated barycentric map provides a predictive mapping from source states to target states. From a machine-learning perspective, this map acts as a structured model of the intervention response, and regularization plays the role of an inductive bias that controls model complexity.

In the following, we show how Schatten OT can reduce the effective rank of couplings and barycentric projection maps. We use four perturbations from the 4i dataset within the CellOT data of Bunne et al. (2023). In particular, we follow Chen et al. (2025) and consider learning regularized couplings from the 4i perturbation data. The processed dataset is publicly available[2]. More details on our algorithmic setup for this experiment are given in Appendix D.

In the experiment of Chen et al. (2025), the authors fit a displacement-sparse neural OT to 4i perturbation data. In the experiment, we see that dimensionality is reduced, but the error is higher, and the method has high variance because it requires fitting an input convex neural network (ICNN). In contrast, in our work, we focus on the recovery of low-rank transport structures, which Chen et al. (2025) do not target. While our experiments here focus on the compression-fidelity tradeoff within the OT objective itself, they motivate Schatten OT as a practical tool for interpretable modeling in high-dimensional biological ML pipelines.

In Figures 4 and 5, we plot the effective rank against $\lambda$ for two different affine maps $\mathcal{A}(\boldsymbol{P}) = \boldsymbol{P}$ and $\mathcal{A}(\boldsymbol{P}) = \boldsymbol{Y}\boldsymbol{P}^\top\boldsymbol{A}^{-1/2}$. We display the result for four different perturbations. For each, we average over five random subsamples of size 500 from the control and perturbation distributions. Since we aim to recover a low-rank coupling and low-rank barycentric projection, we compare against Sinkhorn and Low-Rank Sinkhorn Factorization (Cuturi, 2013; Scetbon et al., 2021). We do not compare against Subspace Elastic OT (Klein et al., 2024) since it is not designed to recover a low-rank coupling or low-rank barycentric map. In each figure, we display both effective rank versus regularization and cost ratio versus regularization. The cost ratio for an estimated $\hat{\boldsymbol{P}}$ is $\langle\boldsymbol{C},\hat{\boldsymbol{P}}\rangle/\langle\boldsymbol{C},\boldsymbol{P}_{\mathsf{Sink},1}\rangle$, where $\boldsymbol{P}_{\mathsf{Sink},1}$ is the solution obtained from Sinkhorn with regularization parameter 1. We compare against this rather than the OT cost for stability.

In Figure 4, we display the result for low-rank coupling recovery. As we can see, we can drastically reduce the complexity of transport plans without paying much more in transport costs. On the other hand, while Sinkhorn and Low-Rank Sinkhorn reduce the rank, they more dramatically increase the transport cost. Next, in Figure 5, we repeat the previous experiment, focusing on recovering a low-rank barycentric projection map. We see again that Schatten-1 regularization can reduce the effective rank, though the transport cost increases more substantially. However, transport cost increases less substantially for Schatten OT than for Sinkhorn and Low-Rank Sinkhorn.

---

[2]https://www.research-collection.ethz.ch/handle/20.500.11850/609681

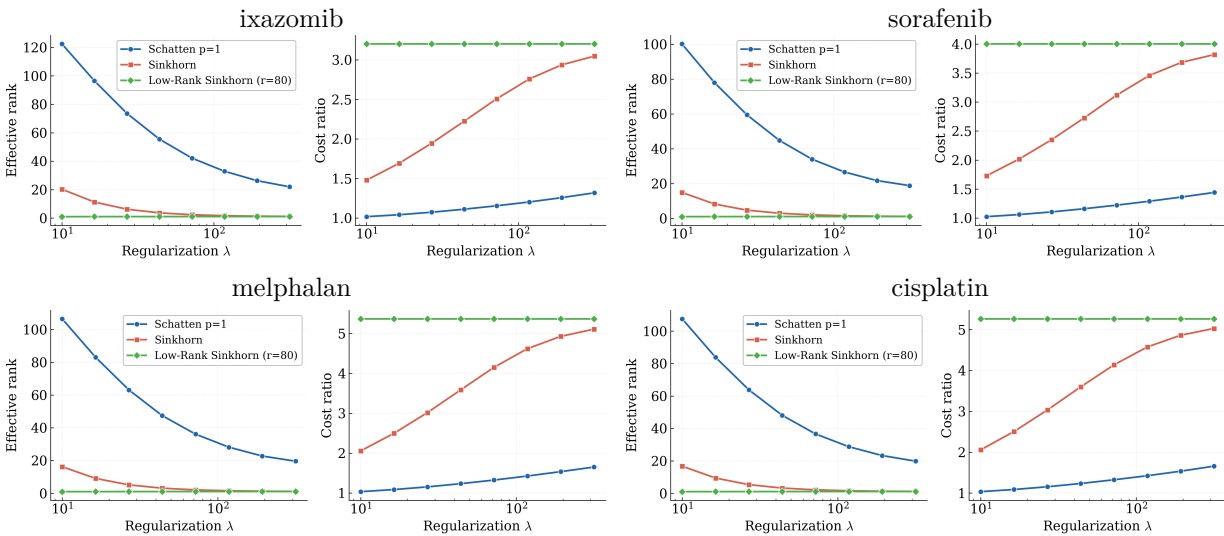

Figure 4: Plots of the performance of regularized OT methods for low-rank coupling recovery across four perturbations from the 4i perturbation data of Bunne et al. (2023). We compare Sinkhorn, Schatten OT with $p = 1$, and Low-Rank Sinkhorn Factorization. For each perturbation, we plot the effective rank of each method's coupling versus the regularization parameter, as well as the cost ratio compared to Sinkhorn with parameter 1. As we can see, Schatten OT can reduce the effective rank without increasing the transport cost too much.

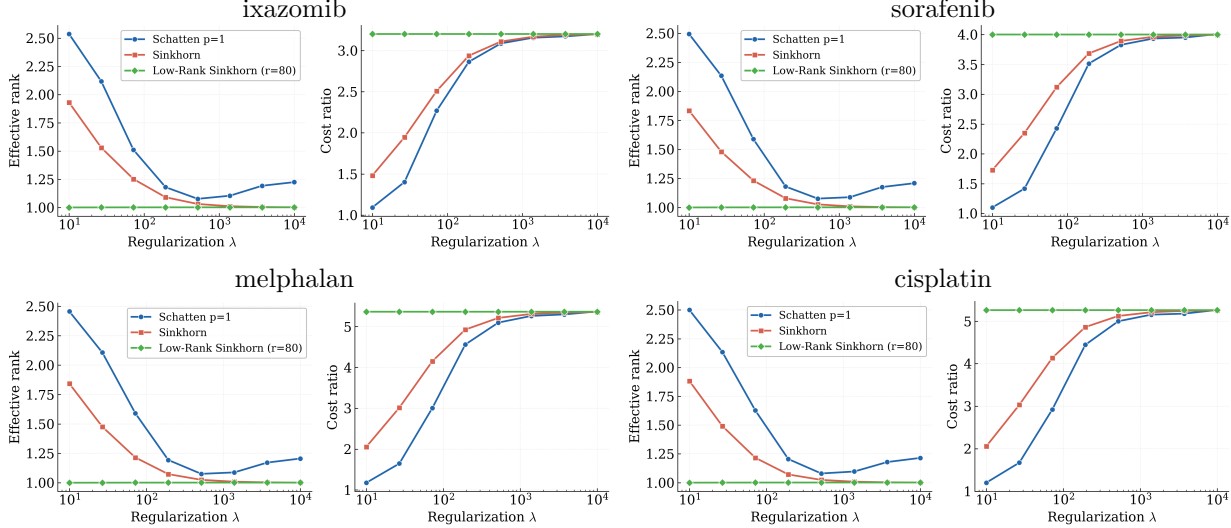

Figure 5: Plots of the performance of regularized OT methods in low-rank barycentric map recovery on the 4i perturbation data of Bunne et al. (2023). The setup is the same as Figure 4, except now we seek to recover a low-rank barycentric map rather than a low-rank coupling. From these figures, we see that Schatten OT can reduce the effective rank of this map, though the transport cost now increases more. On the other hand, while Sinkhorn also reduces the rank of this map, it incurs a more drastic increase in the relative transport cost. Low-Rank Sinkhorn is not stable in this experiment.

# 6 Conclusion

We introduced Schatten-$p$ regularized OT (Schatten OT), a unified convex framework for incorporating low-dimensional structure into OT problems. A key advantage of our formulation lies in its convexity and

generality. Convexity allows us, for the first time, to provide provable recovery results in illustrative yet straightforward examples. Generality allows us to penalize any affine function of the coupling, thereby simultaneously encompassing many existing OT regularizations and enabling new ones.

Theoretically, we established the first recovery guarantees for low-rank couplings and low-rank barycentric displacements, bridging ideas from compressed sensing and OT theory. Algorithmically, we developed an efficient mirror descent method to solve these regularized problems in practice. Empirically, this approach performs well and demonstrates practical utility on 4i cell-perturbation data. Our results show that Schatten OT recovers low-rank structure with only modest increases in transport cost, yielding simpler and more interpretable transport maps.

We believe this work paves the way for more interpretable and scalable OT methods. In particular, the Schatten OT framework may provide a foundation for connecting OT to broader advances in sparse modeling, compressed sensing, and interpretable machine learning.

### Limitations

While Schatten OT provides a convex mechanism for promoting low-rank transport structure and can be solved efficiently with first-order methods, several limitations remain. First, the recovery guarantees in Section 4 are proved in a stylized setting and should be viewed as qualitative guidance rather than a full characterization of when low-rank structure will emerge in practice. Second, although our mirror descent solver avoids explicit semidefinite formulations, each iteration requires computing a nuclear-norm subgradient, which entails an SVD of $\mathcal{A}(\boldsymbol{P})$ and can become a computational bottleneck at large scale. In large problems, truncated or randomized SVD variants are likely necessary. Third, we do not perform an exhaustive empirical study, and we leave a systematic empirical evaluation across regularizations and formulations to future work. Finally, while OT is a general-purpose tool for machine learning practice, we do not conduct an in-depth study of the advantages of this new methodology for specific machine learning tasks. Connections to real machine learning applications, where one can consider learned or task-specific OT models, may further clarify the regimes where Schatten OT offers the greatest practical benefit.

### Acknowledgments

T. Maunu was supported by the NSF under Award No. 2305315.

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

## A   Extension to the Continuous Setting

Up until now, we have focused our attention on formulations in the discrete case. However, there is a direct extension of Schatten OT to the continuous setting by taking Schatten-$p$ norms of appropriate linear operators over general Hilbert spaces. In this section, we let $\mu, \nu \in \mathcal{P}_2(\mathbb{R}^d)$ be general measures. The set of couplings between these measures is $\Pi(\mu, \nu)$.

To define our extension to the continuous case, we let $\mathcal{A} : \mathcal{P}_2(\mathbb{R}^d \times \mathbb{R}^d) \to \mathcal{B}(\mathcal{H})$ be a map from the space of couplings to the set of bounded linear operators on some Hilbert space $\mathcal{H}$. Let $\| \cdot \|_{S_p}$ now denote the Schatten-$p$ norm over $\mathcal{B}(\mathcal{H})$, which is defined as $\|T\|_{S_p}^p = \text{Tr}[(T^*T)^{p/2}]$. Then, we define the continuous Schatten OT problem

$$\text{Sch-OT}_p(\mu, \nu; (\lambda, p, q, \mathcal{A})) := \min_{\pi \in \Pi(\mu, \nu)} \mathbb{E}_{(X,Y)\sim\pi} \|X - Y\|^2 + \lambda \|\mathcal{A}(\pi)\|_{S_p}^q. \tag{9}$$

We note that, as in the discrete case, this notion depends on choices of $\lambda$, $p$, $q$, and $\mathcal{A}$. As before, choosing $p, q \geq 1$ and $\mathcal{A}$ an affine map makes the problem (9) convex.

Below, we give some examples of affine maps that extend our discrete examples. Throughout, we let $\rho = \mu \otimes \nu$ be the reference product measure.

**Covariance regularization:**   The most direct connection between the continuous and discrete cases is to penalize moments of the distribution. In the continuous case, this corresponds to regularizing the covariance of $\pi$. In this case, all of the regularizations discussed in Section 2.3 are the same except we regularize the linear operator over $\mathbb{R}^d$, $\Sigma_\pi = \mathbb{E}_\pi \begin{pmatrix} X \\ Y \end{pmatrix} \begin{pmatrix} X \\ Y \end{pmatrix}^\top$.

**Continuous sparse and low-rank regularization:**   We now discuss the analogs of quadratic and low-rank regularization, the coupling matrix $P$. Here, we can take $\mathcal{A}(\pi) = S_\pi : L^2(\nu) \to L^2(\mu)$ as the linear operator

$$(S_\pi f)(\boldsymbol{x}) = \int f(y) \frac{d\pi}{d\rho} d\nu(y) = \mathbb{E}_\pi[f(Y)|X = \boldsymbol{x}].$$

Note that $S_\pi$ is affine in $\pi$. Then, regularizing the Schatten-$p$ norm of $S_\pi$ corresponds to using the Schatten-$p$ norm of $P$ as discussed earlier. The continuous quadratic case has already been studied by Lorenz et al. (2021), though not framed as a Schatten-2 norm of $S_\pi$.

**Elastic costs:**   We can recover the elastic costs by taking the Schatten norm of a specifically constructed operator. Choose a measurable partition of $\mathbb{R}^d \times \mathbb{R}^d$ given by $(E_k)_{k\in\mathbb{N}}$ with $\rho(E_k) > 0$. Define an orthonormal family in $L^2(\rho)$ by

$$\phi_k(\boldsymbol{x}, \boldsymbol{y}) = \frac{\mathbb{1}((\boldsymbol{x}, \boldsymbol{y}) \in E_k)}{\sqrt{\rho(E_k)}},$$

where $\mathbb{1} : \mathbb{R}^d \times \mathbb{R}^d \to \{0, 1\}$ is the indicator function of a set. Set the diagonal weights to be $s_k(\pi) = \int_{E_k} \|\boldsymbol{y} - \boldsymbol{x}\|_1 d\pi$, which are linear in $\pi$. Then, letting $(e_k)_{k \in \mathbb{N}}$ be a basis for $\ell_2$, we can define the linear operator $\mathcal{A}(\pi) : \ell^2 \to L^2(\rho)$ by

$$A(\pi)e_k = s_k(\pi)\phi_k.$$

Notice that this is again linear in $\pi$, and furthermore $A(\pi)e_k$ are orthogonal with $\|A(\pi)e_k\| = s_k(\pi)$. Therefore, the singular values of $\mathcal{A}(\pi)$ are $s_k(\pi)$. Thus

$$\|\mathcal{A}(\pi)\|_{S_1} = \sum_k s_k(\pi) = \int \|\boldsymbol{y} - \boldsymbol{x}\|_1 d\pi.$$

A similar construction yields the subspace elastic costs discussed in Section 2.3. The general principle here is that elastic OT problems can be embedded as Schatten OT regularized problems over appropriate operators.

**Barycentric projection maps and displacements:** We can also consider Schatten-$p$ regularization of the barycentric projection maps and displacements. In the continuous case, the barycentric projection map $T_\pi : \mathbb{R}^d \to \mathbb{R}^d$ is $T_\pi(x) = \mathbb{E}_\pi[Y|X = x]$. Let the displacement map be defined as $D_\pi(x) = T_\pi(x) - x$. Then, we can formulate a Schatten-$p$ penalization of this barycentric displacement map by viewing $T_\pi$ or $D_\pi$ as operators $T_\pi, D_\pi : \mathbb{R}^d \to L^2(\mu)$ given by $T_\pi v = \langle v, T_\pi(\cdot)\rangle$ and $D_\pi v = \langle v, D_\pi(\cdot)\rangle$.

## B Supplementary Proofs

### B.1 Proof of Theorem 4

*Proof.* Define, for each $t$, the cluster indicator mass vectors

$$\boldsymbol{\alpha}^{(t)} \in \mathbb{R}^m, (\boldsymbol{\alpha}^{(t)})_i = \begin{cases} \frac{1}{Rg}, & i \in S_t, \\ 0, & \text{otherwise}, \end{cases}$$

and similarly

$$\boldsymbol{\beta}^{(t)} \in \mathbb{R}^n, (\boldsymbol{\beta}^{(t)})_j = \begin{cases} \frac{1}{Rg}, & j \in T_t, \\ 0, & \text{otherwise}. \end{cases}$$

The rank $R$ coupling we wish to recover is

$$\boldsymbol{P}^\star := \sum_{t=1}^R \boldsymbol{\alpha}^{(t)}\boldsymbol{\beta}^{(t)\top}. \tag{10}$$

Notice that $\boldsymbol{P}^\star \in \mathcal{U}(\boldsymbol{a}, \boldsymbol{b})$ is block-diagonal with blocks $(S_t \times T_t)$ that are uniform (each entry equals $1/(Rg)^2$), and we can explicitly compute $\|\boldsymbol{P}^\star\|_{S_1} = \frac{1}{R}$. It will be convenient to define the normalized indicator vectors $\boldsymbol{u}^{(t)} = \boldsymbol{\alpha}^{(t)}/\|\boldsymbol{\alpha}^{(t)}\|_2$, $\boldsymbol{v}^{(t)} = \boldsymbol{\beta}^{(t)}/\|\boldsymbol{\beta}^{(t)}\|_2$, which we stack into matrices $\boldsymbol{U}^\star = [\boldsymbol{u}^{(1)}, \dots, \boldsymbol{u}^{(R)}]$, $\boldsymbol{V}^\star = [\boldsymbol{v}^{(1)}, \dots, \boldsymbol{v}^{(R)}]$. In this way, the canonical subgradient of $\|\cdot\|_{S_1}$ at $\boldsymbol{P}^\star$ is $\boldsymbol{G}^\star := \boldsymbol{U}^\star\boldsymbol{V}^{\star\top}$.

**Step 1: Lower bound on $\Delta_{\min}$.** For any $\boldsymbol{x} \in B(\boldsymbol{c}_t, \rho)$, $\boldsymbol{y}_{\text{in}} \in B(\boldsymbol{d}_t, \rho)$, $\boldsymbol{y}_{\text{out}} \in B(\boldsymbol{d}_s, \rho)$ with $s \neq t$,

$$\|\boldsymbol{x} - \boldsymbol{y}_{\text{out}}\| \geq \|\boldsymbol{c}_t - \boldsymbol{d}_s\| - \|\boldsymbol{x} - \boldsymbol{c}_t\| - \|\boldsymbol{y}_{\text{out}} - \boldsymbol{d}_s\| \geq \Gamma - 2\rho,$$

and

$$\|\boldsymbol{x} - \boldsymbol{y}_{\text{in}}\| \leq \|\boldsymbol{c}_t - \boldsymbol{d}_t\| + \|\boldsymbol{x} - \boldsymbol{c}_t\| + \|\boldsymbol{y}_{\text{in}} - \boldsymbol{d}_t\| \leq \gamma + 2\rho.$$

Thus

$$\|\boldsymbol{x} - \boldsymbol{y}_{\text{out}}\|^2 - \|\boldsymbol{x} - \boldsymbol{y}_{\text{in}}\|^2 \geq (\Gamma - 2\rho)^2 - (\gamma + 2\rho)^2,$$

which is positive when (5) holds.

**Step 2: Across-block exclusion via tilted cost.** By convexity of $\|\cdot\|_{S_1}$,

$$\|\boldsymbol{P}\|_{S_1} \geq \|\boldsymbol{P}^\star\|_{S_1} + \langle \boldsymbol{G}^\star, \boldsymbol{P} - \boldsymbol{P}^\star\rangle, \boldsymbol{G}^\star \in \partial\|\boldsymbol{P}^\star\|_{S_1}, \ \boldsymbol{G}^\star = \boldsymbol{U}^\star\boldsymbol{V}^{\star\top}.$$

Hence, for any feasible $P$,

$$\langle \boldsymbol{C}, \boldsymbol{P} \rangle + \lambda \|\boldsymbol{P}\|_{S_1} - (\langle \boldsymbol{C}, \boldsymbol{P}^\star \rangle + \lambda \|\boldsymbol{P}^\star\|_{S_1}) \geq \langle \boldsymbol{S}(\lambda, \boldsymbol{G}^\star), \boldsymbol{P} - \boldsymbol{P}^\star \rangle.$$

For $i \in S_t$ and $j \in T_s$ with $s \neq t$, one has $\boldsymbol{G}^\star_{ij} = 0$, since the left and right singular vectors are block-supported and orthonormal across clusters. On the other hand, for $j' \in T_t$,

$$\boldsymbol{G}^\star_{ij'} = \langle \boldsymbol{u}^{(t)}, \boldsymbol{e}_i \rangle \langle \boldsymbol{v}^{(t)}, \boldsymbol{e}_{j'} \rangle = \frac{a_i}{\|\boldsymbol{\alpha}^{(t)}\|_2} \cdot \frac{b_{j'}}{\|\boldsymbol{\beta}^{(t)}\|_2} = \frac{1}{g}.$$

Therefore, for any $i \in S_t$, $s \neq t$, $j \in T_s$, and $j' \in T_t$,

$$\boldsymbol{S}_{ij}(\lambda, \boldsymbol{G}^\star) - \boldsymbol{S}_{ij'}(\lambda, \boldsymbol{G}^\star) = (\|\boldsymbol{x}_i - \boldsymbol{y}_j\|_2^2 - \|\boldsymbol{x}_i - \boldsymbol{y}_{j'}\|_2^2) - \lambda \cdot \frac{1}{g} \geq \Delta_{\min} - \frac{\lambda}{g}.$$

If $\lambda < g\Delta_{\min}$, these gaps are strictly positive, so no $S(\lambda, \boldsymbol{G}^\star)$-optimal coupling can place mass across clusters. Any minimizer of the original problem must then be block-supported on $\bigcup_t(S_t \times T_t)$.

**Step 3: Within-block tie-breaking via the nuclear norm.** By the distance equality condition in Assumption 3, all within-cluster couplings have equal transport cost. Fix $t$. We can restrict any feasible coupling $\boldsymbol{P} \in \mathcal{U}(\boldsymbol{a}, \boldsymbol{b})$ to the block $(S_t, T_t)$, which we denote as $\boldsymbol{P}_{S_t, T_t}$. We note that this can be written as

$$\boldsymbol{P}_{S_t, T_t} = \mathbf{1}_g \mathbf{1}_g^\top / g^2 + \boldsymbol{M}^{(t)}, \text{ where } \boldsymbol{M}^{(t)} \mathbf{1} = \mathbf{0}, (\boldsymbol{M}^{(t)})^\top \mathbf{1} = \mathbf{0}.$$

In other words, we can represent it as rank-1 product coupling plus a perturbation with $\mathbf{0}$ row/column sums. Choose an orthonormal basis of $\mathbb{R}^g$ on the target side with the first vector proportional to $\mathbf{1}_g$. Then $\boldsymbol{M}^{(t)}$ lives entirely in the orthogonal complement of $\mathbf{1}_g$. The standard inequality $\|\cdot\|_{S_1} \geq \|\cdot\|_{S_2}$ yields

$$\|\boldsymbol{P}_{S_t, T_t}\|_{S_1} \geq \|\boldsymbol{P}_{S_t, T_t}\|_{S_2} = \sqrt{\|\mathbf{1}_g \mathbf{1}_g^\top / g^2\|_{S_2}^2 + \|\boldsymbol{M}^{(t)}\|_{S_2}^2} > \|\mathbf{1}_g \mathbf{1}_g^\top / g^2\|_{S_2} = \|\mathbf{1}_g/g\|_2 \|\mathbf{1}_g/g\|_2,$$

whenever $\boldsymbol{M}^{(t)} \neq 0$. Summing over $t$ shows that among all block-supported couplings, the nuclear norm is uniquely minimized at $\boldsymbol{M}^{(t)} \equiv 0$, i.e., at the uniform block $\boldsymbol{P}^\star$.

Combining these three steps proves the theorem. $\qquad \square$

## B.2 Proof of Theorem 6

*Proof.* We proceed in four steps. We will show that the coupling we recover, $\boldsymbol{P}^\star \in \mathcal{U}(\boldsymbol{a}, \boldsymbol{b})$, satisfies the condition $\boldsymbol{P}^\star_{i,(t,+)} = \boldsymbol{P}^\star_{i,(t,-)} = \frac{1}{2}a_i$ if $i \in S_t$ otherwise $\boldsymbol{P}^\star_{i,(s,\sigma)} = 0$ if $s \neq t$ or $i \notin S_t$.

**Step 1: Feasibility and rank-$1$ structure.** By construction, $\boldsymbol{P}^\star \in \mathcal{U}(\boldsymbol{a}, \boldsymbol{b})$ and, for each $i \in S_t$,

$$T_{\boldsymbol{P}^\star}(\boldsymbol{x}_i) = \tfrac{1}{2}(\boldsymbol{y}_{t,+} + \boldsymbol{y}_{t,-}) = \boldsymbol{m}_t.$$

Hence $T_{\boldsymbol{P}^\star}(\boldsymbol{x}_i) - \boldsymbol{x}_i = \boldsymbol{m}_t - \boldsymbol{x}_i = -\xi_i \boldsymbol{u}$ when $\boldsymbol{x}_i = \boldsymbol{m}_t + \xi_i \boldsymbol{u}$ for $|\xi_i| \leq \rho$, which is true by assumption. Writing $\boldsymbol{\gamma} \in \mathbb{R}^m$ such that $\gamma_i = -\xi_i \sqrt{a_i}$, we have

$$\mathcal{A}(\boldsymbol{P}^\star) = \boldsymbol{u}\boldsymbol{\gamma}^\top.$$

Therefore, $\text{rank}\mathcal{A}(\boldsymbol{P}^\star) = 1$, and $\|\mathcal{A}(\boldsymbol{P}^\star)\|_{S_1} = \|\boldsymbol{\gamma}\|_2$.

**Step 2: A tilted-cost lower bound and across-cluster margin.** Let $\boldsymbol{G}^\star$ be a canonical subgradient of the nuclear norm at $\boldsymbol{B}^\star := \mathcal{A}(\boldsymbol{P}^\star)$:

$$\boldsymbol{G}^\star \in \partial \|\boldsymbol{B}^\star\|_{S_1}, \boldsymbol{G}^\star = \boldsymbol{u}\boldsymbol{w}^\top, \text{ where } \boldsymbol{w} := \frac{\boldsymbol{\gamma}}{\|\boldsymbol{\gamma}\|_2}.$$

For any $\boldsymbol{P} \in \mathcal{U}(\boldsymbol{a}, \boldsymbol{b})$, by convexity of the nuclear norm,

$$\|\mathcal{A}(\boldsymbol{P})\|_{S_1} \geq \|\boldsymbol{B}^\star\|_{S_1} + \langle \boldsymbol{G}^\star, \mathcal{A}(\boldsymbol{P}) - \boldsymbol{B}^\star \rangle. \tag{11}$$

Using $\mathcal{A}(\boldsymbol{P}) - \boldsymbol{B}^\star = \boldsymbol{Y}(\boldsymbol{P} - \boldsymbol{P}^\star)^\top \boldsymbol{A}^{-1/2}$ and cyclicity of the trace,

$$\langle \boldsymbol{G}^\star, \mathcal{A}(\boldsymbol{P}) - \boldsymbol{B}^\star \rangle = \langle \boldsymbol{A}^{-1/2} \boldsymbol{G}^{\star\top} \boldsymbol{Y}, \boldsymbol{P} - \boldsymbol{P}^\star \rangle. \tag{12}$$

Combining (11) and (12) with the objective $F_\lambda(\boldsymbol{P}) := \langle \boldsymbol{C}, \boldsymbol{P} \rangle + \lambda \|\mathcal{A}(\boldsymbol{P})\|_{S_1}$ and the definition of the tilted cost $\boldsymbol{S}(\lambda, \boldsymbol{G})$ in (4) yields the lower bound

$$F_\lambda(\boldsymbol{P}) - F_\lambda(\boldsymbol{P}^\star) \geq \langle \boldsymbol{S}(\lambda, \boldsymbol{G}^\star), \boldsymbol{P} - \boldsymbol{P}^\star \rangle. \tag{13}$$

We can compute the tilted costs $\boldsymbol{S}(\lambda, \boldsymbol{G}^\star)$ explicitly: for any $i$ and $(t, \sigma)$,

$$(\boldsymbol{A}^{-1/2} \boldsymbol{G}^{\star\top} \boldsymbol{Y})_{i,(t,\sigma)} = \frac{1}{\sqrt{a_i}} w_i \langle \boldsymbol{u}, \boldsymbol{y}_{t,\sigma} \rangle = \frac{\gamma_i \mu_t}{\|\boldsymbol{\gamma}\|_2 \sqrt{a_i}} = -\frac{\xi_i \mu_t}{\|\boldsymbol{\gamma}\|_2}.$$

Therefore, assuming that $i \in S_t$ and for any $s \neq t$, $\sigma \in \{\pm\}$,

$$\boldsymbol{S}_{i,(s,\sigma)}(\lambda, \boldsymbol{G}^\star) - \boldsymbol{S}_{i,(t,\pm)}(\lambda, \boldsymbol{G}^\star) = \underbrace{\|x_i - y_{s,\sigma}\|^2 - \|x_i - y_{t,\pm}\|^2}_{=\Delta_{i,s}} + \lambda \left( -\frac{\xi_i}{\|\boldsymbol{\gamma}\|_2} \right) (\mu_s - \mu_t) \tag{14}$$

By assumption,

$$\Delta_{i,s} \geq |\mu_t - \mu_s|(|\mu_t - \mu_s| - 2\rho) \geq \Lambda(\Lambda - 2\rho) > 0. \tag{15}$$

Also, since $|\xi_i| \leq \rho$, we can bound $\|\boldsymbol{\gamma}\|_2^2 = \sum_{k=1}^m a_k \xi_k^2 \leq \rho^2 \sum_{k=1}^m a_k = \rho^2$. This implies that $\frac{|\xi_i|}{\|\boldsymbol{\gamma}\|_2} \leq 1$. Thus, we can extend the lower bound in (14)

$$\boldsymbol{S}_{i,(s,\sigma)}(\lambda, \boldsymbol{G}^\star) - \boldsymbol{S}_{i,(t,\pm)}(\lambda, \boldsymbol{G}^\star) \geq \Lambda((\Lambda - 2\rho) - \lambda). \tag{16}$$

Thus, at $\boldsymbol{G}^\star$, for every $\lambda \in [0, \Lambda - 2\rho)$, across-cluster tilted costs are strictly greater than within-cluster tilted costs. Therefore any tilted cost optimal coupling must match $\boldsymbol{x}_i$ to $\{\boldsymbol{y}_{t,\pm}\}$ for $i \in S_t$.

**Step 3: Within-cluster degeneracy and the nuclear-norm tie-break.** Fix $t \in [R]$. For $i \in S_t$, any within-cluster move between the symmetric targets $(t, +)$ and $(t, -)$ has zero cost difference,

$$\|\boldsymbol{x}_i - \boldsymbol{y}_{t,+}\|^2 = \|\boldsymbol{x}_i - \boldsymbol{y}_{t,-}\|^2.$$

Moreover, the tilted cost is the same for $(t, +)$ and $(t, -)$, since $\langle \boldsymbol{u}, \boldsymbol{y}_{t,+} \rangle = \langle \boldsymbol{u}, \boldsymbol{y}_{t,-} \rangle = \mu_t$. Therefore, for any feasible $\boldsymbol{P}$ that sends mass within clusters (i.e., $\text{supp}(\boldsymbol{P}) \subseteq \{(i, (t, \pm)) : i \in S_t\}$),

$$\langle \boldsymbol{S}(\lambda, \boldsymbol{G}^\star), \boldsymbol{P} - \boldsymbol{P}^\star \rangle = 0. \tag{17}$$

For such $\boldsymbol{P}$, we can write the within-cluster mass split by $p_i \in [0, 1]$ so that

$$\boldsymbol{P}_{i,(t,+)} = p_i a_i, \quad \boldsymbol{P}_{i,(t,-)} = (1 - p_i) a_i$$

A direct computation gives

$$T_{\boldsymbol{P}}(\boldsymbol{x}_i) = p_i \boldsymbol{y}_{t,+} + (1 - p_i) \boldsymbol{y}_{t,-} = \boldsymbol{m}_t + (2p_i - 1)\varepsilon \boldsymbol{v}, \varepsilon \geq 0.$$

Hence, with

$$\boldsymbol{\beta} \in \mathbb{R}^n, \ \beta_i := (2p_i - 1)\varepsilon \sqrt{a_i},$$

we obtain the rank-$\leq 2$ decomposition

$$\mathcal{A}(\boldsymbol{P}) = \boldsymbol{u}\boldsymbol{\gamma}^\top + \boldsymbol{v}\boldsymbol{\beta}^\top. \tag{18}$$

We claim that, for any $\boldsymbol{\beta} \neq 0$,

$$\|\boldsymbol{u}\boldsymbol{\gamma}^\top + \boldsymbol{v}\boldsymbol{\beta}^\top\|_{S_1} > \|\boldsymbol{u}\boldsymbol{\gamma}^\top\|_{S_1} = \|\boldsymbol{\gamma}\|_2. \tag{19}$$

Indeed, let $\boldsymbol{Q} \in \mathbb{R}^{d \times d}$ be an orthogonal matrix whose first two columns are $\boldsymbol{u}$ and $\boldsymbol{v}$. Orthogonal invariance of singular values implies

$$\|\boldsymbol{u}\boldsymbol{\gamma}^\top + \boldsymbol{v}\boldsymbol{\beta}^\top\|_{S_1} = \|\begin{bmatrix} \boldsymbol{\gamma} & \boldsymbol{\beta} & \boldsymbol{0} & \cdots & \boldsymbol{0} \end{bmatrix}\|_{S_1} = \sigma_1 + \sigma_2,$$

where $\sigma_1 \geq \sigma_2 \geq 0$. If $\boldsymbol{\beta}$ is not colinear with $\boldsymbol{\gamma}$, the matrix has rank 2, so $\sigma_2 > 0$, and $\sigma_1 \geq \|\boldsymbol{\gamma}\|_2$ (since $\|\boldsymbol{M}\|_2 \geq$ the Euclidean norm of any row). Hence $\sigma_1 + \sigma_2 > \|\boldsymbol{\gamma}\|_2$. If instead $\boldsymbol{\beta} = c\boldsymbol{\gamma}$ for some $c \neq 0$, then the matrix has rank 1 with singular value $\sqrt{\|\boldsymbol{\gamma}\|_2^2 + \|\boldsymbol{\beta}\|_2^2} = \sqrt{1 + c^2}\|\boldsymbol{\gamma}\|_2 > \|\boldsymbol{\gamma}\|_2$. Thus (19) holds in all cases $\beta \neq 0$.

Combining (13) and (17), for any within-cluster feasible $P$,

$$F_\lambda(\boldsymbol{P}) - F_\lambda(\boldsymbol{P}^\star) \geq \lambda(\|\mathcal{A}(\boldsymbol{P})\|_{S_1} - \|\boldsymbol{B}^\star\|_{S_1}) = \lambda(\|\boldsymbol{u}\boldsymbol{\gamma}^\top + \boldsymbol{v}\boldsymbol{\beta}^\top\|_{S_1} - \|\boldsymbol{\gamma}\|_2), \tag{20}$$

which is strictly positive by (19) whenever $\beta \neq 0$, i.e., whenever some $p_i \neq \frac{1}{2}$.

**Step 4: Optimality and uniqueness for $\lambda \in [0, \Lambda - 2\rho)$.** Let $\lambda \in [0, \Lambda - 2\rho)$. For any feasible $\boldsymbol{P}$, decompose $\boldsymbol{P} - \boldsymbol{P}^\star$ into an across-cluster part and a within-cluster part. By (16),

$$\langle S(\lambda, \boldsymbol{G}^\star), \boldsymbol{P} - \boldsymbol{P}^\star \rangle > 0$$

if $\boldsymbol{P}$ sends any mass across clusters, and (13) implies $F_\lambda(\boldsymbol{P}) > F_\lambda(\boldsymbol{P}^\star)$ in that case. Therefore, any minimizer of $F_\lambda$ must be supported within clusters. For within-cluster couplings, (20) implies $F_\lambda(\boldsymbol{P}) > F_\lambda(\boldsymbol{P}^\star)$ unless $p_i = \frac{1}{2}$ for all $i$, i.e., if $\boldsymbol{P} = \boldsymbol{P}^\star$. Consequently, $\boldsymbol{P}^\star$ is the *unique* minimizer of the Schatten OT problem for $\lambda \in [0, \Lambda - 2\rho)$. $\qquad\square$

## C  The Gaussian Case

The previous section treated recovery of low-rank structures in discrete OT. We now discuss an application of the continuous Schatten regularization (9) for Gaussians.

We treat two Gaussian specializations of the Schatten-$p$ programs discussed earlier: (i) a nuclear-norm penalty that promotes low-rank cross-covariance, and (ii) a nuclear-norm penalty that promotes low-rank transport. As emphasized in our general framework, the barycentric projection $x \mapsto \mathbb{E}_\pi[Y \mid X = x]$ is an *affine* map of the coupling $\pi$, so the induced Schatten-$p$ penalty is convex in $\pi$ for $p \geq 1$. The same holds for Schatten penalties applied to any affine image $A(\pi)$.

For simplicity, we consider the mean zero case. Let $\mu = \mathcal{N}(\mathbf{0}, \boldsymbol{\Sigma}_0)$ and $\nu = \mathcal{N}(\mathbf{0}, \boldsymbol{\Sigma}_1)$ on $\mathbb{R}^d$ with $\boldsymbol{\Sigma}_0, \boldsymbol{\Sigma}_1 \succ 0$. A *Gaussian coupling* is a joint Gaussian $\pi = \mathcal{N}\left(\begin{bmatrix} \mathbf{0} \\ \mathbf{0} \end{bmatrix}, \begin{bmatrix} \boldsymbol{\Sigma}_0 & \boldsymbol{K} \\ \boldsymbol{K}^\top & \boldsymbol{\Sigma}_1 \end{bmatrix}\right)$, parameterized by a cross-covariance $\boldsymbol{K} \in \mathbb{R}^{d \times d}$ satisfying the feasibility constraint

$$\begin{bmatrix} \boldsymbol{\Sigma}_0 & \boldsymbol{K} \\ \boldsymbol{K}^\top & \boldsymbol{\Sigma}_1 \end{bmatrix} \succeq 0 \iff \|\boldsymbol{\Sigma}_0^{-1/2} \boldsymbol{K} \boldsymbol{\Sigma}_1^{-1/2}\|_2 \leq 1. \tag{21}$$

We denote the set of all such $\boldsymbol{K}$ as $\mathcal{K}(\boldsymbol{\Sigma}_0, \boldsymbol{\Sigma}_1)$. Equivalently, we denote the set of all Gaussian couplings between $\mu$ and $\nu$ as $\Pi_g(\mu, \nu)$.

For the quadratic cost $c(\boldsymbol{x}, \boldsymbol{y}) = \|\boldsymbol{x} - \boldsymbol{y}\|^2$, the transport cost under $\pi$ is $\mathbb{E}_\pi \|X - Y\|^2 = \mathrm{tr}(\boldsymbol{\Sigma}_0) + \mathrm{tr}(\boldsymbol{\Sigma}_1) - 2\mathrm{tr}(\boldsymbol{K})$. Moreover the barycentric map induced by $\pi$ is

$$T_\pi(\boldsymbol{x}) = \boldsymbol{A}_\pi \boldsymbol{x}, \ \boldsymbol{A}_\pi := \boldsymbol{K}^\top \boldsymbol{\Sigma}_0^{-1}. \tag{22}$$

### C.1  Gaussian Low-rank Cross-Covariance

Consider the Gaussian Schatten OT problem

$$\min_{\boldsymbol{K} \in \mathcal{K}(\boldsymbol{\Sigma}_0, \boldsymbol{\Sigma}_1)} \mathrm{tr}(\boldsymbol{\Sigma}_0) + \mathrm{tr}(\boldsymbol{\Sigma}_1) - 2\mathrm{tr}(\boldsymbol{K}) + \lambda\|\boldsymbol{K}\|_{S_1}. \tag{23}$$

This is a semidefinite program.

We can solve this problem in closed form. Let $\boldsymbol{S} := \boldsymbol{\Sigma}_1^{1/2}\boldsymbol{\Sigma}_0^{1/2}$ and write its SVD $\boldsymbol{S} = \boldsymbol{U}\mathrm{diag}(\sigma_1,\ldots,\sigma_d)\boldsymbol{V}^\top$ with $\sigma_1 \geq \cdots \geq \sigma_d > 0$. Feasible $\boldsymbol{K}$ can be written as $\boldsymbol{K} = \boldsymbol{\Sigma}_0^{1/2}\boldsymbol{M}\boldsymbol{\Sigma}_1^{1/2}$ with $\|\boldsymbol{M}\|_2 \leq 1$. By von Neumann's trace inequality, $\mathrm{tr}(\boldsymbol{K}) = \mathrm{tr}(\boldsymbol{S}\boldsymbol{M}) \leq \sum_i \sigma_i s_i$ where $s_i$ are the singular values of $\boldsymbol{M}$, with equality when $\boldsymbol{M}$ shares singular vectors with $\boldsymbol{S}$. At optimum we may thus take $\boldsymbol{M} = \boldsymbol{U}\mathrm{diag}(s_1,\ldots,s_d)\boldsymbol{V}^\top$ with $0 \leq s_i \leq 1$ and $\|\boldsymbol{K}\|_{S_1} = \|\boldsymbol{M}\|_{S_1} = \sum_i s_i$. The objective in (23) reduces (up to constants) to

$$\min_{0 \leq s_i \leq 1} \sum_{i=1}^{d} (\lambda - 2\sigma_i)s_i = \sum_{i=1}^{d} \min_{0 \leq s \leq 1} (\lambda - 2\sigma_i)s,$$

which is separable and linear in each $s_i$. Hence, we arrive at the following proposition.

**Proposition 7** (Hard spectral selection)**.** *The unique minimizer of* (23) *is obtained by* hard thresholding *the singular spectrum of $\boldsymbol{S}$:*

$$s_i^\star = \mathbf{1}\{\sigma_i > \lambda/2\}, \qquad \boldsymbol{K}_\lambda = \boldsymbol{\Sigma}_0^{1/2}\boldsymbol{U}\mathrm{diag}(s_i^\star)\boldsymbol{V}^\top\boldsymbol{\Sigma}_1^{1/2}. \tag{24}$$

*In particular,* $\mathrm{rank}(\boldsymbol{K}_\lambda) = \#\{i : \sigma_i > \lambda/2\}$.

We include examples to illustrate this hard-thresholding rule.

(i) *Isotropic case:* $\boldsymbol{\Sigma}_0 = a^2\boldsymbol{I}, \boldsymbol{\Sigma}_1 = b^2\boldsymbol{I}$ gives $\boldsymbol{S} = ab\boldsymbol{I}$ and hence either $\boldsymbol{K}_\lambda = ab\boldsymbol{I}$ if $\lambda < 2ab$, or $\boldsymbol{K}_\lambda = 0$ if $\lambda \geq 2ab$.

(ii) *Commuting covariances:* If $\boldsymbol{\Sigma}_0 = \boldsymbol{U}\mathrm{diag}(\boldsymbol{a})\boldsymbol{U}^\top$ and $\boldsymbol{\Sigma}_1 = \boldsymbol{U}\mathrm{diag}(\boldsymbol{b})\boldsymbol{U}^\top$, then $\sigma_i = \sqrt{a_i b_i}$ and $\boldsymbol{K}_\lambda = \boldsymbol{U}\mathrm{diag}(\sqrt{\boldsymbol{a} \odot \boldsymbol{b}}\mathbf{1}\{\sqrt{\boldsymbol{a} \odot \boldsymbol{b}} > \lambda/2\})\boldsymbol{U}^\top$. Here, $\odot$ is the Hadamard (elementwise) product.

We note that the inclusion of a Schatten-2 penalty in this program results in soft thresholding.

## C.2   Gaussian Barycentric Displacements

We now penalize the (weighted) barycentric displacement. In this case, the resulting convex program is

$$\min_{\pi \in \Pi_g(\mu,\nu)} \mathrm{tr}(\boldsymbol{\Sigma}_0) + \mathrm{tr}(\boldsymbol{\Sigma}_1) - 2\mathrm{tr}(\boldsymbol{K}) + \lambda\big\|(\boldsymbol{A}_\pi - \boldsymbol{I})\boldsymbol{\Sigma}_0^{1/2}\big\|_{S_1}, \tag{25}$$

with $\boldsymbol{A}_\pi$ as in (22).

We can give a closed form when $\boldsymbol{\Sigma}_0$ and $\boldsymbol{\Sigma}_1$ commute, which already reveals a clear structure. Suppose there exists an orthogonal $\boldsymbol{U}$ with $\boldsymbol{\Sigma}_0 = \boldsymbol{U}\mathrm{diag}(\boldsymbol{a})\boldsymbol{U}^\top$ and $\boldsymbol{\Sigma}_1 = \boldsymbol{U}\mathrm{diag}(\boldsymbol{b})\boldsymbol{U}^\top$. Any feasible $\boldsymbol{K}$ aligned with $\boldsymbol{U}$ takes the form $\boldsymbol{K} = \boldsymbol{U}\mathrm{diag}(\sqrt{\boldsymbol{a} \odot \boldsymbol{b}} \odot \boldsymbol{m})\boldsymbol{U}^\top$ with $0 \leq m_i \leq 1$. Then $\boldsymbol{A}_\pi = \boldsymbol{K}^\top\boldsymbol{\Sigma}_0^{-1}$ is diagonal in the same basis with entries

$$\alpha_i := \frac{\sqrt{a_i}m_i\sqrt{b_i}}{a_i} = m_i\sqrt{\frac{b_i}{a_i}},$$

which implies that the diagonal values of $(\boldsymbol{A}_\pi - I)\boldsymbol{\Sigma}_0^{1/2}$ are $m_i\sqrt{b_i} - \sqrt{a_i}$ in this basis as well. Hence $\|(\boldsymbol{A}_\pi - \boldsymbol{I})\boldsymbol{\Sigma}_0^{1/2}\|_{S_1} = \sum_i |m_i\sqrt{b_i} - \sqrt{a_i}|$, and

$$\mathrm{tr}(\boldsymbol{K}) = \sum_i \sqrt{a_i}m_i\sqrt{b_i}.$$

Consequently, (25) decouples into $d$ scalar problems over $m_i \in [0,1]$:

$$\min_{0 \leq m \leq 1} \phi_{\boldsymbol{a},\boldsymbol{b},\lambda}(m) := -2\sqrt{ab}m + \lambda|m\sqrt{b} - \sqrt{a}|. \tag{26}$$

**Theorem 8.** *Fix $(\boldsymbol{a}, \boldsymbol{b}, \lambda)$ with $\boldsymbol{a}, \boldsymbol{b} > 0$. The unique minimizer of* (26) *is*

$$
m^\star = \begin{cases} 1, & \text{if } b \leq a, \\ 1, & \text{if } b > a \text{ and } \lambda < 2\sqrt{a}, \\ \sqrt{a/b}, & \text{if } b > a \text{ and } \lambda \geq 2\sqrt{a}. \end{cases}
$$

*Equivalently, the coupling that solves* (25) *has barycentric projection map $T_\pi(\cdot) = \boldsymbol{A}_\lambda \cdot$, where*

$$
\boldsymbol{A}_\lambda = \boldsymbol{U}\,\mathrm{diag}(\boldsymbol{\alpha}^\star)\boldsymbol{U}^\top, \qquad \alpha_i^\star = \begin{cases} \sqrt{b_i/a_i}, & \text{if } b_i \leq a_i, \\ \sqrt{b_i/a_i}, & \text{if } b_i > a_i \text{ and } \lambda < 2\sqrt{a_i}, \\ 1, & \text{if } b_i > a_i \text{ and } \lambda \geq 2\sqrt{a_i}. \end{cases} \tag{27}
$$

The proof follows from analyzing the 1-dimensional optimization problem (26).

In words, the regularizer does not suppress contracting directions ($b_i \leq a_i$), and it prunes expanding directions $b_i > a_i$ back to the identity once $\lambda$ crosses the sharp threshold $2\sqrt{a_i}$. Thus, $\mathrm{rank}(\boldsymbol{A}_\lambda - \boldsymbol{I})$ equals the number of contracting eigendirections plus the number of expanding eigendirections with $\lambda < 2\sqrt{a_i}$.

As an example, consider the isotropic case. If $\boldsymbol{\Sigma}_0 = \sigma_0^2\boldsymbol{I}, \boldsymbol{\Sigma}_1 = \sigma_1^2\boldsymbol{I}$. If $\sigma_1 \leq \sigma_0$, then $\boldsymbol{A}_\lambda = (\sigma_1/\sigma_0)\boldsymbol{I}$ for all $\lambda$. On the other hand, if $\sigma_1 > \sigma_0$, then $\boldsymbol{A}_\lambda = (\sigma_1/\sigma_0)\boldsymbol{I}$ for $\lambda < 2\sigma_0$, and $\boldsymbol{A}_\lambda = I$ for $\lambda \geq 2\sigma_0$.

### C.3 Discussion

In the Gaussian case, for $p > 1$, the programs remain convex. In the commuting case, we can find separable one-dimensional convex problems similar to those we found in the previous sections. For larger $p$, we would observe smooth shrinkage of the spectrum $m_i$ rather than hard thresholds at $p = 1$. We leave a detailed analysis of the noncommuting case to future work.

## D Details on 4i Experiment Setup

For reproducibility, we give more details on the 4i experiment here. We subsample source and target points to form measures with 500 points each. We then run mirror descent with the desired regularization, where the initial step size is $\eta_0 = 0.1$ for low-rank coupling recovery and $\eta_0 = 10^{-4}$ for low-rank barycentric map recovery. A diminishing step size of $\eta_k = \eta_0/\sqrt{k+1}$ is used. We run mirror descent for a maximum of 50 iterations using Sinkhorn projection (with up to 500 Sinkhorn iterations per mirror descent step), unless the marginal error reaches $10^{-12}$. This is followed by the rounding procedure of Altschuler et al. (2017) to ensure iterates remain in the transport polytope. We return the averaged iterate as our proposed solution to the Schatten OT problem.

