# OpenReview forum: "Simplifying Optimal Transport through Schatten-$p$ Regularization"
_TMLR — Accepted by TMLR_

### Review · Reviewer_aTux · 2025-10-26

**Summary Of Contributions:**

This paper introduces "Schatten OT," a novel and general framework for finding low-rank structures in Optimal Transport by using Schatten-p norm regularization. Its primary contribution is providing a single, convex optimization program (for $p, q \ge 1$) that unifies a wide range of existing methods, such as those for finding low-rank couplings, sparse plans, and elastic costs. A major strength is that this convexity enables, for the first time, provable theoretical guarantees for recovering low-rank solutions in OT, a significant finding that connects OT with compressed sensing principles. The authors also propose a scalable mirror-descent algorithm and provide strong empirical evidence on synthetic and real 4i cell data, showing the method effectively simplifies transport plans with only a modest increase in cost. Key weaknesses include the fact that the novel recovery theorems are currently limited to simplified "toy" settings and that the framework, as presented, does not encompass the widely-used entropic regularization.

**Audience:**

Yes

**Audience Explanation:**

The paper directly tackles two of the most significant challenges in computational OT: scalability and interpretability. The paper introduces a unified, convex framework. This convexity allows the authors to provide the first provable recovery guarantees for low-rank solutions in OT, a significant theoretical contribution that would interest the research community. It develops a scalable mirror-descent algorithm to solve the new formulation and demonstrates its effectiveness on real-world 4i cell-perturbation data.

**Broader Impact Concerns:**

The paper is mostly theoretical, so there are no broader impact concerns.

**Claims And Evidence:**

Yes

**Claims Explanation:**

- **Claim**: The paper proposes a general, convex framework called Schatten OT that unifies many existing regularization methods (like low-rank couplings, quadratic regularization, and elastic costs) and enables new ones.
- The framework is clearly defined in Equation (3). The authors then explicitly demonstrate how this single equation can recover several existing methods (low-rank couplings, quadratic regularization, group sparsity, elastic costs) simply by changing the affine map $\mathcal{A}$ and the parameters $p$ and $q$.  They also show how it creates new regularizers for barycentric maps and covariances.
- **Claim**: The framework's convexity (for $p,q \ge 1$) allows for direct theoretical analysis and provable recovery guarantees for low-rank solutions.
- Theorem 4: Proves exact recovery of a low-rank, block-diagonal coupling in a specific clustered setting. Theorem 6: Proves exact recovery of a rank-1 barycentric displacement map in a symmetric two-cluster setting.
- **Claim**: The proposed mirror descent algorithm efficiently solves the Schatten OT problem and performs well on real data, recovering low-rank structures with only a modest increase in transport cost.
- Fig. 1 & 2 clearly show that as the regularization strength $\lambda$ increases, the "effective rank" of the solution decreases significantly. Fig. 3 directly supports the algorithmic claims, showing sublinear convergence. Fig. 4 experiment demonstrates practical utility.

**Requested Changes:**

- Justification for the choice of mirror descent over other potential convex solvers, such as ADMM or primal-dual methods, would also strengthen the algorithmic contribution by providing clearer context
- A brief discussion or even a simple experiment on combining the Schatten-p penalty with an entropic regularizer, since (3) cannot recover entropic regularization.

---

> ### Author Response · Authors · 2025-11-24
> **Response Part 1/2**
>
> We thank the reviewer for the careful reading and the constructive comments. We are glad that you found the framework, theory, and experiments convincing, and we appreciate your suggestions for clarifying the algorithmic choices and the relation to entropic regularization. Below, we address your two requested changes in detail and describe the corresponding modifications we will make to the manuscript.
>
> *“Justification for the choice of mirror descent over other potential convex solvers, such as ADMM or primal-dual methods, would also strengthen the algorithmic contribution by providing clearer context.”*
>
> We agree that it is helpful to better motivate the choice of mirror descent with KL geometry, and we will expand the discussion in Section 3 accordingly. In particular, we will include a subsection on the choice of algorithm and outline potential alternatives based on the following points:
>
> 1. First, we want to emphasize that the core computational bottleneck in solving (3) at scale is repeatedly enforcing the marginal constraints, i.e., projecting onto the transportation polytope. In Euclidean geometry, this projection is expensive, whereas in KL geometry it coincides with Sinkhorn scaling, for which there now exist highly optimized and near–linear-time implementations. Using KL mirror descent, therefore, lets us reuse the same projection primitive that underlies large‑scale entropic OT solvers, but for a much more general convex objective.
> 2. Each mirror descent iteration consists of: one SVD to compute the (sub)gradient of the Schatten term, and one Sinkhorn projection to return to the transport polytope. This clean separation makes it easy to plug Schatten OT into existing OT codebases: the only nonstandard operation is the SVD, while the OT machinery (Sinkhorn + rounding) is unchanged. By contrast, ADMM or primal-dual splitting would introduce auxiliary variables and dual constraints, making it harder to isolate and reuse the OT projection structure.
> 4. KL mirror descent automatically preserves nonnegativity and is well aligned with the simplex‑like geometry of the transport polytope. ADMM or Euclidean primal–dual schemes would either require explicit nonnegativity projections or log‑barrier style treatments, which complicates the implementation and can add additional regularization.
> 4. In the convex regime, KL mirror descent yields standard $O(1/\sqrt{T})$ convergence guarantees with minimal assumptions. The algorithmic pseudocode is short and self‑contained, which we view as an advantage for reproducibility. While more sophisticated primal–dual or ADMM variants may be competitive or even superior in some regimes, carefully tuning and analyzing them for the general Schatten OT objective is nontrivial, and we see this as an interesting avenue for follow‑up work rather than a prerequisite for the present paper.
> 5. One advantage we see of the primal-dual setting is that, for certain regularizers within the Schatten OT framework, one can avoid the explicit full SVD computation. However, this does not extend to all Schatten OT formulations and is problem-dependent. We will highlight which formulations have simple implementations in the revision (e.g., when $p=q=2$ and $\mathcal{A}(P)=P$, group-sparse Frobenius norm on blocks of $P$, elastic cost formulations). On the other hand, primal-only versions of the problem, like mirror descent, do not always need to use a full SVD (e.g., when $p=1$, we can use the top singular subspace).

---

> ### Author Response · Authors · 2025-11-24
> **Response Part 2/2**
>
> *“A brief discussion or even a simple experiment on combining the Schatten-p penalty with an entropic regularizer, since (3) cannot recover entropic regularization.”*
>
> You are correct that SchattenOT by itself does not encompass classical entropic regularization, since the negative entropy is not a Schatten norm of an affine map of the coupling $P$. We agree that it is important to clarify how entropic and Schatten‑$p$ penalties can be combined, and what this implies algorithmically. In the current version, we mention this only briefly; we will expand the discussion and make the combined formulation explicit.
>
> The main points we address are the following. First, from an algorithmic perspective, our mirror descent scheme extends naturally to this setting:
> - The KL geometry is unchanged, and the projection step is still implemented via Sinkhorn scaling.
> - The entropic term adds a simple contribution to the gradient in the mirror‑descent update.
> - Thus, one can view the resulting algorithm as a “Schatten‑regularized Sinkhorn” method that simultaneously encourages low‑rank (or more generally low‑dimensional) structure via the Schatten terms and smooths the coupling via entropy.
>
> This leads to a two‑parameter family interpolating smoothly between standard entropic OT and the purely Schatten‑regularized regime. We expect this combined formulation to be beneficial when one wishes to retain the numerical stability of entropic OT while still promoting structural simplicity in the coupling or its induced maps.
>
> Because our current experiments already focus on illustrating the effect of Schatten regularization on the rank and cost of the solutions, and because the combined model introduces an additional tuning dimension, we have not added a full experimental grid over both parameters in the present submission. We have instead chosen to provide a more detailed conceptual discussion, and we now make explicit that exploring this combined entropic–Schatten regime is an interesting direction for future empirical work.
>
> We will expand the discussion at the end of Section 2.3 as follows:
>
> **While many past regularizations for OT fall into this framework, some do not. In particular, classical entropic regularization cannot be recovered from (3), because the negative entropy is not a Schatten norm of an affine function of $P$. Nevertheless, entropy can be combined with Schatten OT in a straightforward way by considering objectives with both the Schatten OT and entropic regularizers. For convex Schatten regularization , this problem remains convex and can be solved by the same KL mirror-descent algorithm: the entropic term contributes an additional term to the gradient, while the KL projection onto the transport polytope is still implemented via Sinkhorn scaling. In this way, we can interpolate between standard entropic OT and the purely Schatten-regularized regime studied in this work. A systematic empirical study of this combined entropic–Schatten formulation is an interesting direction for future work.**

---

> > ### Comment · Reviewer_aTux · 2025-12-20
> >
> > Thanks for your response, My concerns have been addressed. I recommend accepting the paper.

---

### Review · Reviewer_Aqgm · 2025-11-07

**Summary Of Contributions:**

In the manuscript, the authors introduce Schatten-p Regularized Optimal Transport (Schatten OT), a unified convex framework that generalizes many existing regularized OT methods including low-rank, sparse, and quadratic formulations under a single mathematical model. By leveraging ideas from compressed sensing, the authors establish the first theoretical recovery guarantees for low-rank couplings and barycentric displacements in OT. They also develop an efficient mirror-descent algorithm with convergence guarantees for solving the convex formulation at scale. Empirical results on synthetic and biological datasets demonstrate that Schatten OT effectively recovers low-rank, interpretable transport maps with minimal cost increases, thereby bridging OT theory, convex optimization, and compressed sensing in a principled and scalable manner.

**Audience:**

Yes

**Audience Explanation:**

Yes. The paper’s findings would interest TMLR’s audience, particularly researchers in machine learning, optimization, and computational optimal transport, as it introduces a novel convex framework with theoretical guarantees and scalable algorithms for low-rank and interpretable transport.

**Broader Impact Concerns:**

There are no major ethical or broader concerns associated with this work.

**Claims And Evidence:**

Yes

**Claims Explanation:**

Yes. The paper’s claims are well supported by clear theoretical, algorithmic, and empirical evidence. The authors rigorously derive convexity and optimality conditions for the proposed Schatten OT framework and provide formal theorems with proofs establishing low-rank recovery guarantees which is an important novelty in the OT literature. The mirror-descent algorithm is mathematically justified with convergence analysis, and experiments on both synthetic and real-world (4i perturbation) datasets convincingly demonstrate the method’s efficiency, scalability, and ability to recover interpretable low-rank structures. Together, the theoretical results and empirical validations align coherently with the stated contributions and meet TMLR’s standards for clarity, soundness, and reproducibility.

**Requested Changes:**

While the paper is very well written and contains many strengths, the following changes could make its findings more robust and broadly impactful.

1. Clarify Practical Impact (Important): I believe the theoretical results from the paper are strong, but the paper would benefit from a clearer discussion of how Schatten OT could be applied in real-world problems like single-cell analysis, imaging, or generative modeling and when the low-rank assumption is most justified.
2. Expanded Empirical Evaluation (Important): The experiments are convincing but could be strengthened by including comparisons with more recent or diverse OT baselines, such as neural or elastic regularized OT models. This would help demonstrate where Schatten OT provides the most tangible improvements in practice.
3. Hyperparameter Sensitivity (Minor): Including a short experiment or discussion showing how the regularization strength (lambda) and Schatten parameter (p) affect results would make the method easier to reproduce and tune for new datasets.
4. Discussion of Limitations and Future Work (Minor): It would be useful for the authors to explicitly mention potential limitations, such as performance in highly nonconvex or noisy settings, and suggest possible future directions.

---

> ### Author Response · Authors · 2025-11-24
> **Response Part 1/2**
>
> We thank the reviewer for the thoughtful and detailed feedback. We are glad that you find the theoretical, algorithmic, and empirical components convincing, and that the contributions are of interest to the TMLR audience. Below, we address your requested changes point by point and describe the edits we will make to the manuscript.
>
> *“The paper would benefit from a clearer discussion of how Schatten OT could be applied in real-world problems like single-cell analysis, imaging, or generative modeling, and when the low-rank assumption is most justified.”*
>
> We agree that making the practical impact more explicit will improve the paper. In the revision, we will add a short subsection that discusses concrete use cases and gives guidance on when a low‑rank (or more generally low‑dimensional) structure is a reasonable modeling assumption.
>
> Concretely, we will discuss:
> - Single-cell/4i perturbation data: We will expand the discussion around the 4i experiment to explain that Schatten OT is particularly well suited to longitudinal or perturbation studies where high‑dimensional measurements (e.g., gene expression or imaging features) vary along a small number of latent programs. In this setting, a low‑rank coupling or low‑rank barycentric displacement naturally corresponds to a small number of “transport archetypes”.
> - Imaging: We will highlight image‑based settings (e.g., registration of related images or domains with coherent deformations) where mass is expected to move along a few spatial or intensity modes.
> - Generative modeling: We will discuss Wasserstein‑based generative models where the data distribution concentrates near a low‑dimensional manifold, and the generator itself is low‑dimensional. In such cases, a locally low‑rank barycentric map is a natural inductive bias.

---

> ### Author Response · Authors · 2025-11-24
> **Response Part 2/2**
>
> *“Experiments could be strengthened by including comparisons with more recent or diverse OT baselines, such as neural or elastic regularized OT models.”*
>
> We agree that including a broader set of baselines will help clarify where Schatten OT sits in the literature. Therefore, we will make the following changes to our experiments by including comparisons to low-rank Sinkhorn factorization for low-rank couplings and subspace elastic OT for low-rank displacements.
> 1. For the low‑rank recovery experiments on synthetic data in Section 5.1, we already compare how the effective rank and transport cost change as we vary the Schatten parameter p and regularization strength lambda. In the revised version, we will additionally include entropic OT (Sinkhorn) as a baseline, showing that its couplings remain high‑rank and dense even when the cost is similar, reinforcing the advantage of Schatten OT for low‑dimensional structure, as well as subspace Elastic OT and low-rank Sinkhorn factorization. For the low-rank transport plan experiment in Figure 1, low-rank Sinkhorn factorization provides a natural comparison, allowing us to vary the enforced rank of the transport plan and the entropic regularization parameter. For the low-rank displacement in Figure 2, Subspace elastic OT allows for recovery of low-rank displacements.
> 2. On the 4i dataset in Section 5.3, we currently compare Schatten OT to an entropic Sinkhorn baseline. In the revision, we will add comparisons to the low-rank Sinkhorn and subspace Elastic OT.
> Taken together, these additions will make it more straightforward to see in which regimes Schatten OT provides tangible differences over standard entropic OT and other low-rank OT variants.
>
> Finally, these other methods that we propose to compare against are nonconvex and lack theoretical guarantees. Thus, we will also emphasize more that this is a distinct advantage of considering the convex program. We will expand the "Discussion" section of the Theorem (Section 4.3) to further emphasize this point.

---

### Review · Reviewer_PDPV · 2025-11-14

**Summary Of Contributions:**

The abstract of the paper correctly summarises it. A convex
formulation for optimal transport is proposed which (like some
existing methods) promotes sparse maps. Since the formulation is
convex some optimality results are derivable. A mirror-descent
algorithm is used since, despite the convex nature of the optimisation
problem, it can get too large for off-the-shelf convex optimisation
algorithms to solve. Empirical results on both synthetic and real data
are presented.

**Additional Comments:**

TMLR is a machine learning journal. The submission makes a brief
argument (by referencing other work) that OT is of use for ML. But
this is basically an OT, not an ML paper. I'm not sure why the authors
have chosen to submit it to an ML journal.

**Audience:**

Yes

**Audience Explanation:**

OT is used in ML, so those aware of its use in ML may find something
of interest here. My guess is that this will be a small proportion of
ML researchers, but since ML is so big these days, this small
proportion could be quite large in absolute terms. Those not already
familiar with OT will not read this paper, I suspect.

**Claims And Evidence:**

Yes

**Claims Explanation:**

Theoretical claims are supported by proofs. Empirical results provide
evidence for the claimed positive features of the presented method.

**Requested Changes:**

The paper would be improved if OT could be "sold" more to an ML audience. Would it be possible to have an ML example in the Experiments section?

I can't think of any other useful changes without going over the 12 pages. Note that the proofs are in appendices.


Note that Wasserstein and Schatten should be capitalised in the bibliography.

---

> ### Author Response · Authors · 2025-11-24
> **Response Part 1/2**
>
> We thank the reviewer for their careful reading of the manuscript and for the constructive comments. We are glad that the theoretical and empirical results were found to be accurate and convincing, and that the work may be of interest to part of the TMLR audience.
>
> Below, we respond to the specific points raised.
>
> *“TMLR is a machine learning journal. The submission makes a brief argument (by referencing other work) that OT is of use for ML. But this is basically an OT, not an ML paper. I'm not sure why the authors have chosen to submit it to an ML journal.”*
>
> We understand the concern about scope and agree that the current version could do a better job of appealing to the machine‑learning community. Our primary motivation for this work is that optimal transport is now a standard building block in many direct machine learning tasks, such as generative modeling, domain adaptation, and representation learning. The two main issues we address, scalability and interpretability via low‑dimensional structure, are central to these applications. We note that the most closely related optimal transport works appear in many recent machine learning venues, to name a few:
> - Scetbon, M., Cuturi, M., & Peyré, G. (2021, July). Low-rank sinkhorn factorization. In International Conference on Machine Learning (pp. 9344-9354). PMLR.
> - Cuturi, M., Klein, M., & Ablin, P. (2023). Monge, Bregman and Occam: Interpretable Optimal Transport in High-Dimensions with Feature-Sparse Maps. In International Conference on Machine Learning (pp. 6671-6682). PMLR
> - Klein, M., Pooladian, A. A., Ablin, P., Ndiaye, E., Niles-Weed, J., & Cuturi, M. (2024). Learning elastic costs to shape monge displacements. Advances in Neural Information Processing Systems, 37, 108542-108565.
> We want to emphasize that this is no accident: applied optimal transport is intimately tied to machine learning.
>
> To help clarify this, we propose to revise the introduction and related‑work sections as follows:
> 1. In the Introduction, we will add a dedicated paragraph that explicitly frames Schatten OT as a potential tool for:
>     - scaling OT‑based ML methods (e.g., OT distances and regularizers used in generative modeling and domain adaptation),
>     - inducing sparse/low‑rank structure in transport maps that can be inspected and used for mechanistic or scientific interpretation (e.g., in single‑cell perturbation analysis and sparse autoencoders).
> 2. We will make it more explicit that our convex formulation may be used as a drop-in replacement for existing regularized OT formulations already used in ML pipelines, with two ML‑oriented advantages:
>     - The unified view that Schatten OT gives covers many existing regularization strategies (e.g., quadratic and elastic regularization) as well as some new ones (transport covariance regularization). We believe that this flexibility allows one to tune the regularization to the problem  at hand, allowing for broad applications.
>     - The provable recovery guarantees for low‑rank structures that we recover are an important step to practical OT theory. In particular, recovery theorems are common in the machine learning literature (see the many works on sparse and low-rank recovery, for instance). Having these theorems in hand allows us to understand what sorts of structures the OT method is able to recover explicitly, rather than just acting as a heuristic.
>
> As a whole, we view the work as sitting at the interface between optimal transport and machine learning: the main technical contributions are in OT and convex optimization, but they are designed for and demonstrated on a real machine‑learning problem in learning transport between high‑dimensional perturbations.

---

> ### Author Response · Authors · 2025-11-24
> **Response Part 2/2**
>
> *“The paper would be improved if OT could be ‘sold’ more to an ML audience. Would it be possible to have an ML example in the Experiments section?”*
>
> As mentioned previously, we do believe that the manuscript could be improved for an ML audience, and as you mention the experiments are a great way to do this. In light of this, we also propose the following:
> - Section 5.3 already builds directly on the 4i perturbation setting of Bunne et al. (2023) and Chen et al. (2025), where neural optimal transport is used to learn single‑cell perturbation responses. In fact, this same sort of experiment is present in many recent optimal transport works, since -omic perturbations can naturally be thought of as transport problems. We believe that this is, in effect, a machine‑learning task: learning a predictive model of how gene expression changes under interventions, using OT‑based objectives and regularizers. To better explain this, we will
>     1. Explicitly introduce this experiment as a machine‑learning application (learning perturbation responses), not just as “real data”. In this experiment, we are learning a map between data distributions.
>     2. Briefly summarize how OT is used as a learning objective in the prior work we follow, and clarify that our contribution is to show that Schatten OT can be used in the same setting to obtain simpler (low‑rank) couplings and barycentric maps while maintaining competitive transport cost.
>     3. Emphasize that this demonstrates Schatten OT in a realistic high‑dimensional ML pipeline, rather than only in synthetic OT toy problems.
>     4. Explain how this could be used in downstream tasks,
> - Across Sections 5.1–5.3, we already measure how Schatten OT trades off transport cost against low‑rank structure. We will revise the section headers and opening sentences to frame these experiments using more ML‑centric language:
>     1. highlighting “representation compression”, “model interpretability”, and “dimensionality reduction of transport maps” as ML‑relevant goals;
>     2. explicitly connecting effective rank reduction to simpler downstream models (e.g., fewer latent factors in perturbation responses).
>
>
>  We are cautious about adding a new ML benchmarks that would require substantial space and effort to carry out, since we believe that as classically defined our “real data” experiment is in fact machine learning. Furthermore, a main goal of our text is not the experiments but instead the fact that this represents the first provable guarantees of recovery of low-dimensional structure in optimal transport. That being said, we are currently in the process of extending the cell perturbation experiments to more examples, and we will try to run some classification/clustering tasks to see how these interact with representations learned by Schatten OT.
>
> *“Note that Wasserstein and Schatten should be capitalised in the bibliography.”*
>
> Thank you for pointing this out. We will correct the capitalization of “Wasserstein” and “Schatten” in all bibliography entries.

---

### Decision · Action_Editor_BfBE · 2026-01-17

**Recommendation:** Accept with minor revision

**Additional Comments:**

Please include the promised changes during rebuttal, especially highlight the relevance of OP to the ML community and the additional numerial experiments

**Audience:**

Yes

**Audience Explanation:**

Yes, optimal transport (OT) has been involved in numerous machine learning models, and efficient computation of interpretable OP solutions is crucial for these models

**Claims And Evidence:**

Yes

**Claims Explanation:**

This paper proposes a Schatten-$p$ norm regularization for optimal transport (OT) that unifies several previous regularization schemes, backed by explicit mathematical derivations. The convexity of the Schatten-$p$ norm (for $p \ge 1$) allows the development of efficient numerical algorithms and recovery theory for regularized OT. To this end, the paper proposes a mirror descent algorithm with provable convergence and establishes recovery results for a couple of concrete examples. The paper also presents numerical results to support its theoretical findings.

While some reviewers were initially concerned about the relatively limited empirical evaluation, they were convinced by the additional experiments the authors proposed to include in their revision. Also, given the paper's strong theoretical contributions, as agreed upon by all three reviewers, the current AE considers this a minor point.